# Unsupervised Representation Learning from Pre-trained Diffusion Probabilistic Models

**Zijian Zhang**[1]     **Zhou Zhao**[1]*     **Zhijie Lin**[2]
[1]Department of Computer Science and Technology, Zhejiang University
[2]Sea AI Lab
{ckczzj,zhaozhou}@zju.edu.cn
linzj@sea.com

## Abstract

Diffusion Probabilistic Models (DPMs) have shown a powerful capacity of generating high-quality image samples. Recently, diffusion autoencoders (Diff-AE) have been proposed to explore DPMs for representation learning via autoencoding. Their key idea is to jointly train an encoder for discovering meaningful representations from images and a conditional DPM as the decoder for reconstructing images. Considering that training DPMs from scratch will take a long time and there have existed numerous pre-trained DPMs, we propose **P**re-trained **D**PM **A**uto**E**ncoding (**PDAE**), a general method to adapt existing pre-trained DPMs to the decoders for image reconstruction, with better training efficiency and performance than Diff-AE. Specifically, we find that the reason that pre-trained DPMs fail to reconstruct an image from its latent variables is due to the information loss of forward process, which causes a gap between their predicted posterior mean and the true one. From this perspective, the classifier-guided sampling method can be explained as computing an extra mean shift to fill the gap, reconstructing the lost class information in samples. These imply that the gap corresponds to the lost information of the image, and we can reconstruct the image by filling the gap. Drawing inspiration from this, we employ a trainable model to predict a mean shift according to encoded representation and train it to fill as much gap as possible, in this way, the encoder is forced to learn as much information as possible from images to help the filling. By reusing a part of network of pre-trained DPMs and redesigning the weighting scheme of diffusion loss, PDAE can learn meaningful representations from images efficiently. Extensive experiments demonstrate the effectiveness, efficiency and flexibility of PDAE. Our implementation is available at https://github.com/ckczzj/PDAE.

## 1 Introduction

Deep generative models such as variational autoencoders (VAEs) [25, 39], generative adversarial networks (GANs) [13], autoregressive models [50, 48], normalizing flows (NFs) [38, 23] and energy-based models (EBMs) [9, 45] have shown remarkable capacity to synthesize striking image samples. Recently, another kind of generative models, Diffusion Probabilistic Models (DPMs) [43, 14] are further developed and becoming popular for their stable training process and state-of-the-art sample quality [8]. Although a large number of degrees of freedom in implementation, the DPMs discussed in this paper will refer exclusively to those trained by the denoising method proposed in DDPMs [14].

Unsupervised representation learning via generative modeling is a popular topic in computer vision. Latent variable generative models, such as GANs and VAEs, are a natural candidate for this, since they inherently involve a latent representation of the data they generate. Likewise, DPMs inherently

---

*Corresponding author.

36th Conference on Neural Information Processing Systems (NeurIPS 2022).

yield latent variables through the forward process. However, these latent variables lack high-level semantic information because they are just a sequence of spatially corrupted images. In light of this, diffusion autoencoders (Diff-AE) [36] explore DPMs for representation learning via autoencoding. Specifically, they employ an encoder for discovering meaningful representations from images and a conditional DPM as the decoder for image reconstruction by taking the encoded representations as input conditions. Diff-AE is competitive with the state-of-the-art model on image reconstruction and capable of various downstream tasks.

Following the paradigm of autoencoders, PDAE aims to adapt existing pre-trained DPMs to the decoders for image reconstruction and benefit from it. Generally, pre-trained DPMs cannot accurately predict the posterior mean of $\boldsymbol{x}_{t-1}$ from $\boldsymbol{x}_t$ in the reverse process due to the information loss of forward process, which results in a gap between their predicted posterior mean and the true one. This is the reason that they fail to reconstruct an image ($\boldsymbol{x}_0$) from its latent variables ($\boldsymbol{x}_t$). From this perspective, the classifier-guided sampling method [8] can be explained as reconstructing the lost class information in samples by shifting the predicted posterior mean with an extra item computed by the gradient of a classifier to fill the gap. Drawing inspiration from this method that uses the prior knowledges (class label) to fill the gap, we aim to inversely extract the knowledges from the gap, i.e., learn representations that can help to fill the gap. In light of this, we employ a novel gradient estimator to predict the mean shift according to encoded representations and train it to fill as much gap as possible, in this way, the encoder is forced to learn as much information as possible from images to help the filling. PDAE follows this principle to build an autoencoder based on pre-trained DPMs. Furthermore, we find that the posterior mean gap in different time stages contain different levels of information, so we redesign the weighting scheme of diffusion loss to encourage the model to learn rich representations efficiently. We also reuse a part of network of pre-trained DPMs to accelerate the convergence of our model. Based on pre-trained DPMs, PDAE only needs less than half of the training time that Diff-AE costs to complete the representation learning but still outperforms Diff-AE. Moreover, PDAE also enables some other interesting features.

## 2 Background

### 2.1 Denoising Diffusion Probabilistic Models

DDPMs [14] employ a forward process that starts from the data distribution $q(\boldsymbol{x}_0)$ and sequentially corrupts it to $\mathcal{N}(\boldsymbol{0}, \mathbf{I})$ with Markov diffusion kernels $q(\boldsymbol{x}_t|\boldsymbol{x}_{t-1})$ defined by a fixed variance schedule $\{\beta_t\}_{t=1}^T$. The process can be expressed by:

$$q(\boldsymbol{x}_t|\boldsymbol{x}_{t-1}) = \mathcal{N}(\boldsymbol{x}_t; \sqrt{1-\beta_t}\boldsymbol{x}_{t-1}, \beta_t\mathbf{I}) \qquad q(\boldsymbol{x}_{1:T}|\boldsymbol{x}_0) = \prod_{t=1}^T q(\boldsymbol{x}_t|\boldsymbol{x}_{t-1}), \qquad (1)$$

where $\{\boldsymbol{x}_t\}_{t=1}^T$ are latent variables of DDPMs. According to the rule of the sum of normally distributed random variables, we can directly sample $\boldsymbol{x}_t$ from $\boldsymbol{x}_0$ for arbitrary $t$ with $q(\boldsymbol{x}_t|\boldsymbol{x}_0) = \mathcal{N}(\boldsymbol{x}_t; \sqrt{\bar{\alpha}_t}\boldsymbol{x}_0, (1-\bar{\alpha}_t)\mathbf{I})$, where $\alpha_t = 1 - \beta_t$ and $\bar{\alpha}_t = \prod_{i=1}^t \alpha_i$.

The reverse (generative) process is defined as another Markov chain parameterized by $\theta$ to describe the same but reverse process, denoising an arbitrary Gaussian noise to a clean data sample:

$$p_\theta(\boldsymbol{x}_{t-1}|\boldsymbol{x}_t) = \mathcal{N}(\boldsymbol{x}_{t-1}; \boldsymbol{\mu}_\theta(\boldsymbol{x}_t, t), \boldsymbol{\Sigma}_\theta(\boldsymbol{x}_t, t)) \qquad p_\theta(\boldsymbol{x}_{0:T}) = p(\boldsymbol{x}_T)\prod_{t=1}^T p_\theta(\boldsymbol{x}_{t-1}|\boldsymbol{x}_t), \qquad (2)$$

where $p(\boldsymbol{x}_T) = \mathcal{N}(\boldsymbol{x}_T; \boldsymbol{0}, \mathbf{I})$. It employs $p_\theta(\boldsymbol{x}_{t-1}|\boldsymbol{x}_t)$ of Gaussian form because the reversal of the diffusion process has the identical functional form as the forward process when $\beta_t$ is small [11, 43]. The generative distribution can be represented as $p_\theta(\boldsymbol{x}_0) = \int p_\theta(\boldsymbol{x}_{0:T})d\boldsymbol{x}_{1:T}$.

Training is performed to maximize the model log likelihood $\int q(\boldsymbol{x}_0)\log p_\theta(\boldsymbol{x}_0)d\boldsymbol{x}_0$ by minimizing the variational upper bound of the negative one. The final objective is derived by some parameterization and simplication [14]:

$$\mathcal{L}_{simple}(\theta) = \mathbb{E}_{\boldsymbol{x}_0,t,\epsilon}\left[\left\|\epsilon - \boldsymbol{\epsilon}_\theta(\sqrt{\bar{\alpha}_t}\boldsymbol{x}_0 + \sqrt{1-\bar{\alpha}_t}\epsilon, t)\right\|^2\right], \qquad (3)$$

where $\boldsymbol{\epsilon}_\theta$ is a function approximator to predict $\epsilon$ from $\boldsymbol{x}_t$.

## 2.2 Denoising Diffusion Implicit Models

DDIMs [44] define a non-Markov forward process that leads to the same training objective with DDPMs, but the corresponding reverse process can be much more flexible and faster to sample from. Specifically, one can sample $\boldsymbol{x}_{t-1}$ from $\boldsymbol{x}_t$ using the $\boldsymbol{\epsilon}_\theta$ of some pre-trained DDPMs via:

$$\boldsymbol{x}_{t-1} = \sqrt{\bar{\alpha}_{t-1}}\left(\frac{\boldsymbol{x}_t - \sqrt{1-\bar{\alpha}_t}\cdot\boldsymbol{\epsilon}_\theta(\boldsymbol{x}_t, t)}{\sqrt{\bar{\alpha}_t}}\right) + \sqrt{1-\bar{\alpha}_{t-1}-\sigma_t^2}\cdot\boldsymbol{\epsilon}_\theta(\boldsymbol{x}_t, t) + \sigma_t\epsilon_t\,, \quad (4)$$

where $\epsilon_t \sim \mathcal{N}(\boldsymbol{0}, \mathbf{I})$ and $\sigma_t$ controls the stochasticity of forward process. The strides greater than 1 are allowed for accelerated sampling. When $\sigma_t = 0$, the generative process becomes deterministic, which is named as DDIMs.

## 2.3 Classifier-guided Sampling Method

Classifier-guided sampling method [43, 46, 8] shows that one can train a classifier $p_\phi(\boldsymbol{y}|\boldsymbol{x}_t)$ on noisy data and use its gradient $\nabla_{\boldsymbol{x}_t} \log p_\phi(\boldsymbol{y}|\boldsymbol{x}_t)$ to guide some pre-trained unconditional DDPM to sample towards specified class $\boldsymbol{y}$. The conditional reverse process can be approximated by a Gaussian similar to that of the unconditional one in Eq.(2), but with a shifted mean:

$$p_{\theta,\phi}(\boldsymbol{x}_{t-1}|\boldsymbol{x}_t, \boldsymbol{y}) \approx \mathcal{N}(\boldsymbol{x}_{t-1}; \boldsymbol{\mu}_\theta(\boldsymbol{x}_t, t) + \boldsymbol{\Sigma}_\theta(\boldsymbol{x}_t, t)\cdot\nabla_{\boldsymbol{x}_t} \log p_\phi(\boldsymbol{y}|\boldsymbol{x}_t)\,, \boldsymbol{\Sigma}_\theta(\boldsymbol{x}_t, t))\,. \quad (5)$$

For deterministic sampling methods like DDIMs, one can use score-based conditioning trick [46, 45] to define a new function approximator for conditional sampling:

$$\hat{\boldsymbol{\epsilon}}_\theta(\boldsymbol{x}_t, t) = \boldsymbol{\epsilon}_\theta(\boldsymbol{x}_t, t) - \sqrt{1-\bar{\alpha}_t}\cdot\nabla_{\boldsymbol{x}_t} \log p_\phi(\boldsymbol{y}|\boldsymbol{x}_t)\,. \quad (6)$$

More generally, any similarity estimator between noisy data and conditions can be applied for guided sampling, such as noisy-CLIP guidance [33, 31].

# 3 Method

## 3.1 Forward Process Posterior Mean Gap

Generally, one will train unconditional and conditional DPMs by respectively learning $p_\theta(\boldsymbol{x}_{t-1}|\boldsymbol{x}_t) = \mathcal{N}(\boldsymbol{x}_{t-1}; \boldsymbol{\mu}_\theta(\boldsymbol{x}_t, t), \boldsymbol{\Sigma}_\theta(\boldsymbol{x}_t, t))$ and $p_\theta(\boldsymbol{x}_{t-1}|\boldsymbol{x}_t, \boldsymbol{y}) = \mathcal{N}(\boldsymbol{x}_{t-1}; \boldsymbol{\mu}_\theta(\boldsymbol{x}_t, \boldsymbol{y}, t), \boldsymbol{\Sigma}_\theta(\boldsymbol{x}_t, \boldsymbol{y}, t))$ to approximate the same forward process posterior $q(\boldsymbol{x}_{t-1}|\boldsymbol{x}_t, \boldsymbol{x}_0) = \mathcal{N}(\boldsymbol{x}_{t-1}; \widetilde{\boldsymbol{\mu}}_t(\boldsymbol{x}_t, \boldsymbol{x}_0), \frac{1-\bar{\alpha}_{t-1}}{1-\bar{\alpha}_t}\beta_t\mathbf{I})$. Here $\boldsymbol{y}$ is some condition that contains some prior knowledges of corresponding $\boldsymbol{x}_0$, such as class label. Assuming that both $\boldsymbol{\Sigma}_\theta$ is set to untrained time dependent constants, under the same experimental settings, the conditional DPMs will reach a lower optimized diffusion loss. The experiment in Figure 1 can prove this fact, which means that $\boldsymbol{\mu}_\theta(\boldsymbol{x}_t, \boldsymbol{y}, t)$ is closer to $\widetilde{\boldsymbol{\mu}}_t(\boldsymbol{x}_t, \boldsymbol{x}_0)$ than $\boldsymbol{\mu}_\theta(\boldsymbol{x}_t, t)$. This implies that there exists a gap between the posterior mean predicted by the unconditional DPMs $\left(\boldsymbol{\mu}_\theta(\boldsymbol{x}_t, t)\right)$ and the true one $\left(\widetilde{\boldsymbol{\mu}}_t(\boldsymbol{x}_t, \boldsymbol{x}_0)\right)$. Essentially, the posterior mean gap is caused by the information loss of forward process so that the reverse process cannot recover it in $\boldsymbol{x}_{t-1}$ only according to $\boldsymbol{x}_t$. If we introduce some knowledges of $\boldsymbol{x}_0$ for DPMs, like $\boldsymbol{y}$ here, the gap will be smaller. The more information of $\boldsymbol{x}_0$ that $\boldsymbol{y}$ contains, the smaller the gap is.

Moreover, according to Eq.(5), the Gaussian mean of classifier-guided conditional reverse process contains an extra shift item compared with that of the unconditional one. From the perspective of posterior mean gap, the mean shift item can partially fill the gap and help the reverse process to reconstruct the lost class information in samples. In theory, if $\boldsymbol{y}$ in Eq.(5) contains all information of $\boldsymbol{x}_0$, the mean shift will fully fill the gap and guide the reverse process to reconstruct $\boldsymbol{x}_0$. On the other hand, if we employ a model to predict mean shift according to our encoded representations $\boldsymbol{z}$ and train it to fill as much gap as possible, the encoder will be forced to learn as much information as possible from $\boldsymbol{x}_0$ to help the filling. The more the gap is filled, the more accurate the mean shift is, the more perfect the reconstruction is, and the more information of $\boldsymbol{x}_0$ that $\boldsymbol{z}$ contains. PDAE follows this principle to build an autoencoder based on pre-trained DPMs.

## 3.2 Unsupervised Representation Learning by Filling the Gap

Following the paradigm of autoencoders, we employ an encoder $\boldsymbol{z} = \boldsymbol{E}_\varphi(\boldsymbol{x}_0)$ for learning compact and meaningful representations from input images and adapt a pre-trained unconditional DPM $p_\theta(\boldsymbol{x}_{t-1}|\boldsymbol{x}_t) = \mathcal{N}(\boldsymbol{x}_{t-1}; \boldsymbol{\mu}_\theta(\boldsymbol{x}_t, t), \boldsymbol{\Sigma}_\theta(\boldsymbol{x}_t, t))$ to the decoder for image reconstruction.

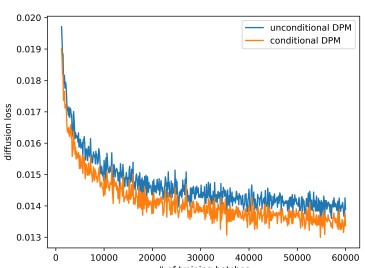

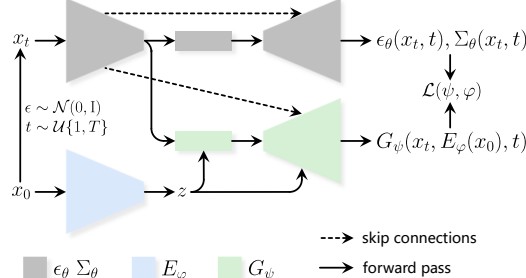

Figure 1: Comparison of diffusion loss between unconditional and conditional DPM trained on MNIST [28].

Figure 2: Network and data flow of PDAE. The gray part represents the pre-trained DPM, which is frozen during training.

Specifically, we employ a gradient estimator $\boldsymbol{G}_\psi(\boldsymbol{x}_t, \boldsymbol{z}, t)$ to simulate $\nabla_{\boldsymbol{x}_t} \log p(\boldsymbol{z}|\boldsymbol{x}_t)$, where $p(\boldsymbol{z}|\boldsymbol{x}_t)$ is some implicit classifier that we will not use explicitly, and use it to assemble a conditional DPM $p_{\theta,\psi}(\boldsymbol{x}_{t-1}|\boldsymbol{x}_t, \boldsymbol{z}) = \mathcal{N}(\boldsymbol{x}_{t-1}; \boldsymbol{\mu}_\theta(\boldsymbol{x}_t, t) + \boldsymbol{\Sigma}_\theta(\boldsymbol{x}_t, t) \cdot \boldsymbol{G}_\psi(\boldsymbol{x}_t, \boldsymbol{z}, t), \boldsymbol{\Sigma}_\theta(\boldsymbol{x}_t, t))$ as the decoder. Then we train it like a regular conditional DPM by optimizing following derived objective (assuming the $\epsilon$-prediction parameterization is adopted):

$$\mathcal{L}(\psi, \varphi) = \mathbb{E}_{\boldsymbol{x}_0, t, \epsilon}\left[\lambda_t \left\| \epsilon - \boldsymbol{\epsilon}_\theta(\boldsymbol{x}_t, t) + \frac{\sqrt{\alpha_t}\sqrt{1 - \bar{\alpha}_t}}{\beta_t} \cdot \boldsymbol{\Sigma}_\theta(\boldsymbol{x}_t, t) \cdot \boldsymbol{G}_\psi(\boldsymbol{x}_t, \boldsymbol{E}_\varphi(\boldsymbol{x}_0), t) \right\|^2\right], \quad (7)$$

where $\boldsymbol{x}_t = \sqrt{\bar{\alpha}_t}\boldsymbol{x}_0 + \sqrt{1 - \bar{\alpha}_t}\epsilon$ and $\lambda_t$ is a new weighting scheme that we will discuss in Section 3.4. Note that we use pre-trained DPMs so that $\theta$ are frozen during the optimization. Usually we set $\boldsymbol{\Sigma}_\theta = \frac{1 - \bar{\alpha}_{t-1}}{1 - \bar{\alpha}_t}\beta_t \mathbf{I}$ to untrained time-dependent constants. The optimization is equivalent to minimizing $\left\| \boldsymbol{\Sigma}_\theta(\boldsymbol{x}_t, t) \cdot \boldsymbol{G}_\psi(\boldsymbol{x}_t, \boldsymbol{E}_\varphi(\boldsymbol{x}_0), t) - (\widetilde{\boldsymbol{\mu}}_t(\boldsymbol{x}_t, \boldsymbol{x}_0) - \boldsymbol{\mu}_\theta(\boldsymbol{x}_t, t)) \right\|^2$, which forces the predicted mean shift $\boldsymbol{\Sigma}_\theta(\boldsymbol{x}_t, t) \cdot \boldsymbol{G}_\psi(\boldsymbol{x}_t, \boldsymbol{E}_\varphi(\boldsymbol{x}_0), t)$ to fill the posterior mean gap $\widetilde{\boldsymbol{\mu}}_t(\boldsymbol{x}_t, \boldsymbol{x}_0) - \boldsymbol{\mu}_\theta(\boldsymbol{x}_t, t)$.

With trained $\boldsymbol{G}_\psi(\boldsymbol{x}_t, \boldsymbol{z}, t)$, we can treat it as the score of an optimal classifier $p(\boldsymbol{z}|\boldsymbol{x}_t)$ and use the classifier-guided sampling method in Eq.(5) for DDPM sampling or use the modified function approximator $\hat{\epsilon}_\theta$ in Eq.(6) for DDIM sampling, based on pre-trained $\boldsymbol{\epsilon}_\theta(\boldsymbol{x}_t, t)$. We put detailed algorithm procedures in Appendix **??**.

Except the semantic latent code $\boldsymbol{z}$, we can infer a stochastic latent code $\boldsymbol{x}_T$ [36] by running the deterministic generative process of DDIMs in reverse:

$$\boldsymbol{x}_{t+1} = \sqrt{\bar{\alpha}_{t+1}}\left(\frac{\boldsymbol{x}_t - \sqrt{1 - \bar{\alpha}_t} \cdot \hat{\boldsymbol{\epsilon}}_\theta(\boldsymbol{x}_t, t)}{\sqrt{\bar{\alpha}_t}}\right) + \sqrt{1 - \bar{\alpha}_{t+1}} \cdot \hat{\boldsymbol{\epsilon}}_\theta(\boldsymbol{x}_t, t). \quad (8)$$

This procedure is optional, but helpful to near-exact reconstruction and real-image manipulation for reconstructing minor details of input images when using DDIM sampling.

We also train a latent DPM $p_\omega(\boldsymbol{z}_{t-1}|\boldsymbol{z}_t)$ to model the learned semantic latent space, same with that in Diff-AE [36]. With a trained latent DPM, we can sample $\boldsymbol{z}$ from it to help pre-trained DPMs to achieve faster and better unconditional sampling under the guidance of $\boldsymbol{G}_\psi(\boldsymbol{x}_t, \boldsymbol{z}, t)$.

### 3.3 Network Design

Figure 2 shows the network and data flow of PDAE. For encoder $\boldsymbol{E}_\varphi$, unlike Diff-AE that uses the encoder part of U-Net [40], we find that simply stacked convolution layers and a linear layer is enough to learn meaningful $\boldsymbol{z}$ from $\boldsymbol{x}_0$. For gradient estimator $\boldsymbol{G}_\psi$, we use U-Net similar to the function approximator $\boldsymbol{\epsilon}_\theta$ of pre-trained DPM. Considering that $\boldsymbol{\epsilon}_\theta$ also takes $\boldsymbol{x}_t$ and $t$ as input, we can further leverage the knowledges of pre-trained DPM by reusing its trained encoder part and time embedding layer, so that we only need to employ new middle blocks, decoder part and output blocks of U-Net for $\boldsymbol{G}_\psi$. To incorporate $\boldsymbol{z}$ into them, we follow [8] to extend Group Normalization [53] by applying scaling & shifting twice on normalized feature maps:

$$\text{AdaGN}(\boldsymbol{h}, t, \boldsymbol{z}) = \boldsymbol{z}_s(t_s\text{GroupNorm}(\boldsymbol{h}) + t_b) + \boldsymbol{z}_b, \quad (9)$$

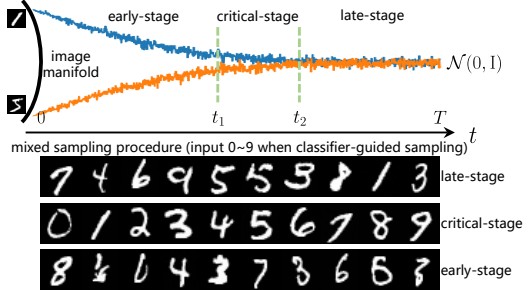
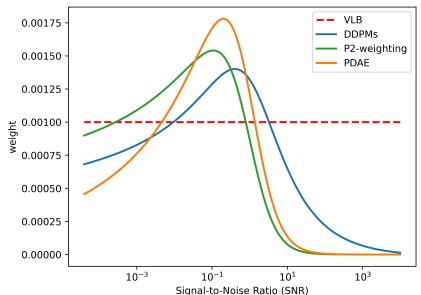

Figure 3: Investigations of the effects of mean shift for different time stages. We perform a 50-step-grid-search for $(t_1, t_2)$ pairs to find the shortest critical-stage that can ensure high accuracy of conditional generation. For MNIST [28], it is $(400, 700)$.

Figure 4: Normalized weighting schemes of diffusion loss for different DPMs relative to the true variational lower bound loss. Linear variance schedule is used.

where $[\boldsymbol{t}_s, \boldsymbol{t}_b]$ and $[\boldsymbol{z}_s, \boldsymbol{z}_b]$ are obtained from a linear projection of $t$ and $\boldsymbol{z}$, respectively. Note that we still use skip connections from reused encoder to new decoder. In this way, $\boldsymbol{G}_\psi$ is totally determined by pre-trained DPM and can be universally applied to different U-Net architectures.

### 3.4 Weighting Scheme Redesign

We originally worked with simplified training objective like that in DDPMs [14], i.e. setting $\lambda_t = 1$ in Eq.(7), but found the training extremely unstable, resulting in slow/non- convergence and poor performance. Inspired by P2-weighting [7], which has shown that the weighting scheme of diffusion loss can greatly affect the performance of DPMs, we attribute this phenomenon to the weighting scheme and investigate it in Figure 3.

Specifically, we train an unconditional DPM and a noisy classifier on MNIST [28], and divide the diffusion forward process into three stages: early-stage between $0$ and $t_1$, critical-stage between $t_1$ and $t_2$ and late-stage between $t_2$ and $T$, as shown in the top row. Then we design a mixed sampling procedure that employs unconditional sampling but switches to classifier-guided sampling only during the specified stage. The bottom three rows show the samples generated by three different mixed sampling procedures, where each row only employs classifier-guided sampling during the specified stage on the right. As we can see, only the samples guided by the classifier during critical-stage match the input class labels. We can conclude that the mean shift during critical-stage contains more crucial information to reconstruct the input class label in samples than the other two stages. From the view of diffusion trajectories, the sampling trajectories are separated from each other during critical-stage and they need the mean shift to guide them towards specified direction, otherwise it will be determined by the stochasticity of Langevin dynamics. Therefore, we opt to down-weight the objective function for the $t$ in early- and late-stage to encourage the model to learn rich representations from critical-stage. Inspired by P2-weighting [7], we redesign a weighting scheme of diffusion loss ($\lambda_t$ in Eq.(7)) in terms of signal-to-noise ratio [24] ($\text{SNR}(t) = \frac{\bar{\alpha}_t}{1 - \bar{\alpha}_t}$):

$$\lambda_t = \big(\frac{1}{1 + \text{SNR}(t)}\big)^{1-\gamma} \cdot \big(\frac{\text{SNR}(t)}{1 + \text{SNR}(t)}\big)^\gamma, \tag{10}$$

where the first item is for early-stage and the second one is for late-stage. $\gamma$ is a hyperparameter that balances the strength of down-weighting between two items. Empirically we set $\gamma = 0.1$. Figure 4 shows the normalized weighting schemes of diffusion loss for different DPMs relative to the true variational lower bound loss. Compared with other DPMs, our weighting scheme down-weights the diffusion loss for both low and high SNR.

## 4 Experiments

To compare PDAE with Diff-AE [36], we follow their experiments with the same settings. Moreover, we also show that PDAE enables some added features. For fair comparison, we use the baseline DPMs provided by official Diff-AE implementation as our pre-trained models (also as our baselines), which

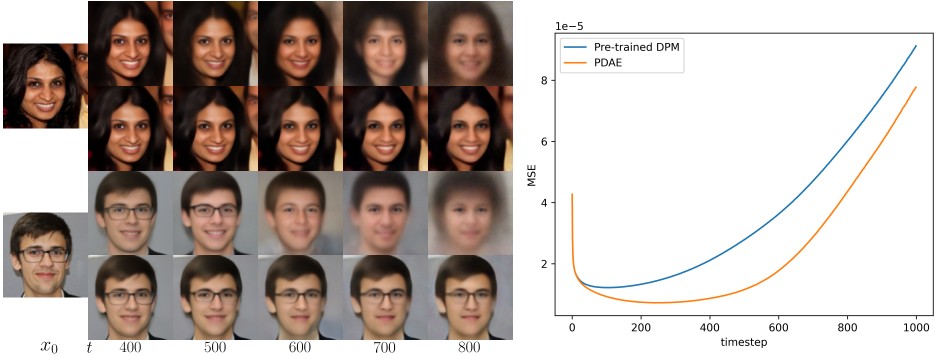

$x_0$    $t$   400      500      600      700      800

Figure 5: **Left**: Predicted $\hat{x}_0$ by denoising $x_t$ for only one step. The first row use pre-trained DPM and the second row use PDAE. **Right**: Average posterior mean gap for all steps.

have the same network architectures (hyperparameters) with their Diff-AE models. For brevity, we use the notation such as "FFHQ128-130M-z512-64M" to name our model, which means that we use a baseline DPM pre-trained with 130M images and leverage it for PDAE training with 64M images, on $128 \times 128$ FFHQ dataset [21], with the semantic latent code $z$ of $512$-$d$. We put all implementation details in Appendix **??** and additional samples of following experiments in Appendix **??**.

## 4.1 Training Efficiency

We demonstrate the better training efficiency of PDAE compared with Diff-AE from two aspects: training time and times. For training time, we train both models with the same network architectures (hyperparameters) on $128 \times 128$ image dataset using $4$ Nvidia A100-SXM4 GPUs for distributed training and set batch size to 128 (32 for each GPU) to calculate their training throughput (imgs/sec./A100). PDAE achieves a throughput of 81.57 and Diff-AE achieves that of 75.41. Owing to the reuse of the U-Net encoder part of pre-trained DPM, PDAE has less trainable parameters and achieves a higher training throughput than Diff-AE. For training times, we find that PDAE needs about $\frac{1}{3} \sim \frac{1}{2}$ of the number of training batches (images) that Diff-AE needs for loss convergence. We think this is because that modeling the posterior mean gap based on pre-trained DPMs is easier than modeling a conditional DPM from scratch. The network reuse and the weighting scheme redesign also help. As a result, based on pre-trained DPMs, PDAE needs less than half of the training time that Diff-AE costs to complete the representation learning.

## 4.2 Learned Mean Shift Fills Posterior Mean Gap

We train a model of "FFHQ128-130M-z512-64M" and show that our learned mean shift can fill the posterior mean gap with qualitative and quantitative results in Figure 5. Specifically, we select some images $x_0$ from FFHQ, sample $x_t = \sqrt{\bar{\alpha}_t}x_0 + \sqrt{1 - \bar{\alpha}_t}\epsilon$ for different $t$ and predict $\hat{x}_0$ from $x_t$ by denoising them for only one step (i.e., $\hat{x}_0 = \frac{x_t - \sqrt{1-\bar{\alpha}_t}\hat{\epsilon}}{\sqrt{\bar{\alpha}_t}}$), using pre-trained DPM and PDAE respectively. As we can see in the figure (left), even for large $t$, PDAE can predict accurate noise from $x_t$ and reconstruct plausible images, which shows that the predicted mean shift fills the posterior mean gap and the learned representation helps to recover the lost information of forward process. Furthermore, we randomly select 1000 images from FFHQ, sample $x_t = \sqrt{\bar{\alpha}_t}x_0 + \sqrt{1 - \bar{\alpha}_t}\epsilon$ and calculate their average posterior mean gap for each step using pre-trained DPM: $\|\widetilde{\mu}_t(x_t, x_0) - \mu_\theta(x_t, t)\|^2$ and PDAE: $\|\widetilde{\mu}_t(x_t, x_0) - (\mu_\theta(x_t, t) + \Sigma_\theta(x_t, t) \cdot G_\psi(x_t, E_\varphi(x_0), t))\|^2$ respectively, shown in the figure (right). As we can see, PDAE predicts the mean shift that significantly fills the posterior mean gap.

## 4.3 Autoencoding Reconstruction

We use "FFHQ128-130M-z512-64M" to run some autoencoding reconstruction examples using PDAE generative process of DDIM and DDPM respectively. As we can see in Figure 6, both methods generate samples with similar contents to the input. Some stochastic variations [36] occur in minor details of hair, eye and skin when introducing stochasticity. Due to the similar performance

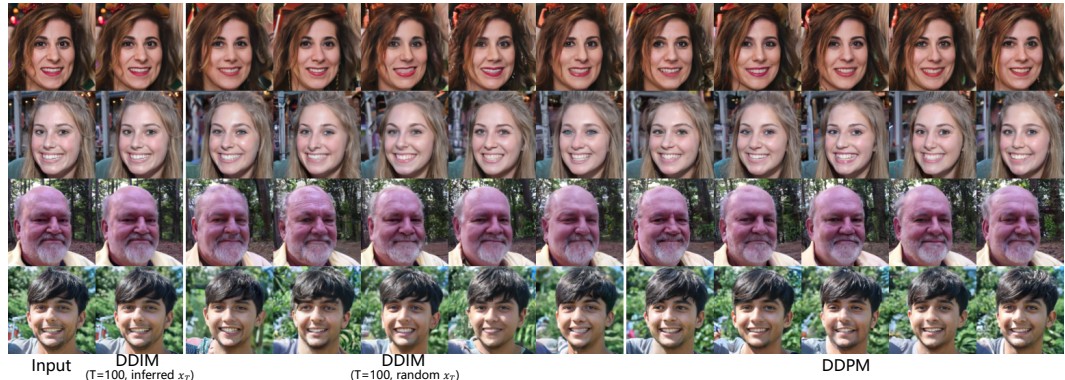

Figure 6: Autoencoding reconstruction examples generated by "FFHQ128-130M-z512-64M" with different sampling methods. Each row corresponds to an example.

Table 1: Autoencoding reconstruction quality of "FFHQ128-130M-z512-64M" on CelebA-HQ.

| Model | Latent dim | SSIM ↑ | LPIPS ↓ | MSE ↓ |
|---|---|---|---|---|
| StyleGAN2 ($\mathcal{W}$ inversion) [22] | 512 | 0.677 | 0.168 | 0.016 |
| StyleGAN2 ($\mathcal{W}+$ inversion) [1, 2] | 7,168 | 0.827 | 0.114 | 0.006 |
| VQ-GAN [10] | 65,536 | 0.782 | 0.109 | 3.61e-3 |
| VQ-VAE2 [37] | 327,680 | 0.947 | 0.012 | 4.87e-4 |
| NVAE [47] | 6,005,760 | 0.984 | **0.001** | **4.85e-5** |
| Diff-AE @130M (T=100, random $\boldsymbol{x}_T$) [36] | 512 | 0.677 | 0.073 | 0.007 |
| PDAE @64M (T=100, random $\boldsymbol{x}_T$) | 512 | 0.696 | 0.094 | 0.005 |
| DDIM @130M (T=100) [44] | 49,152 | 0.917 | 0.063 | 0.002 |
| Diff-AE @130M (T=100, inferred $\boldsymbol{x}_T$) [36] | 49,664 | 0.991 | 0.011 | 6.07e-5 |
| PDAE @64M (T=100, inferred $\boldsymbol{x}_T$) | 49,664 | **0.993** | 0.008 | 5.48e-5 |

between DDPM and DDIM with random $\boldsymbol{x}_T$, we will always use DDIM sampling method in later experiments. We can get a near-exact reconstruction if we use the stochastic latent code inferred from aforementioned ODE, which further proves that the stochastic latent code controls the local details.

To further evaluate the autoencoding reconstruction quality of PDAE, we conduct the same quantitative experiments with Diff-AE. Specifically, we use "FFHQ128-130M-z512-64M" to encode-and-reconstruct all 30k images of CelebA-HQ [20] and evaluate the reconstruction quality with their average SSIM [52], LPIPS [56] and MSE. We use the same baselines described in [36], and the results are shown in Table 1. We can see that PDAE is competitive with the state-of-the-art NVAE even with much less latent dimensionality and also outperforms Diff-AE in all metrics except the LPIPS for random $\boldsymbol{x}_T$. Moreover, PDAE only needs about half of the training times that Diff-AE needs for representation learning, which shows that PDAE can learn richer representations from images more efficiently based on pre-trained DPM.

### 4.4 Interpolation of Semantic Latent Codes and Trajectories

Given two images $\boldsymbol{x}_0^1$ and $\boldsymbol{x}_0^2$ from FFHQ, we use "FFHQ128-130M-z512-64M" to encode them into $(\boldsymbol{z}^1, \boldsymbol{x}_T^1)$ and $(\boldsymbol{z}^2, \boldsymbol{x}_T^2)$ and run PDAE generative process of DDIM starting from $Slerp(\boldsymbol{x}_T^1, \boldsymbol{x}_T^2; \lambda)$ under the guidance of $\boldsymbol{G}_\psi(\boldsymbol{x}_t, Lerp(\boldsymbol{z}^1, \boldsymbol{z}^2; \lambda), t)$ with 100 steps, expecting smooth transitions along $\lambda$. Moreover, from the view of the diffusion trajectories, PDAE generates desired samples by shifting the unconditional sampling trajectories towards the spatial direction predicted by $\boldsymbol{G}_\psi(\boldsymbol{x}_t, \boldsymbol{z}, t)$. This enables PDAE to directly interpolate between two different sampling trajectories. Intuitively, the spatial direction predicted by the linear interpolation of two semantic latent codes, $\boldsymbol{G}_\psi(\boldsymbol{x}_t, Lerp(\boldsymbol{z}^1, \boldsymbol{z}^2; \lambda), t)$, should be equivalent to the linear interpolation of two spatial directions predicted by respective semantic latent code, $Lerp(\boldsymbol{G}_\psi(\boldsymbol{x}_t, \boldsymbol{z}^1, t), \boldsymbol{G}_\psi(\boldsymbol{x}_t, \boldsymbol{z}^2, t); \lambda)$. We present some examples of these two kinds of interpolation methods in Figure 7. As we can see, both methods generate similar samples that smoothly transition from one endpoint to the other, which means that

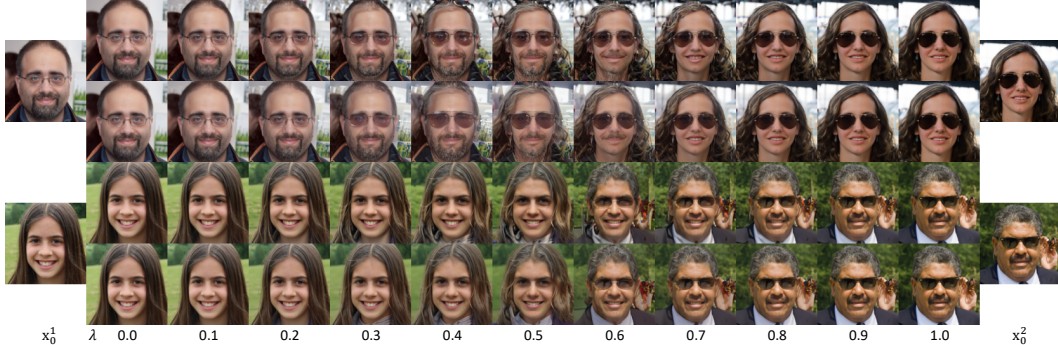

| $x_0^1$ | $\lambda$ | 0.0 | 0.1 | 0.2 | 0.3 | 0.4 | 0.5 | 0.6 | 0.7 | 0.8 | 0.9 | 1.0 | $x_0^2$ |

Figure 7: Interpolation examples generated by "FFHQ128-130M-z512-64M". For each example, the first row use the guidance of $\boldsymbol{G}_\psi\big(\boldsymbol{x}_t, Lerp(\boldsymbol{z}^1, \boldsymbol{z}^2; \lambda), t\big)$ and the second row use the guidance of $Lerp\big(\boldsymbol{G}_\psi(\boldsymbol{x}_t, \boldsymbol{z}^1, t), \boldsymbol{G}_\psi(\boldsymbol{x}_t, \boldsymbol{z}^2, t); \lambda\big)$.

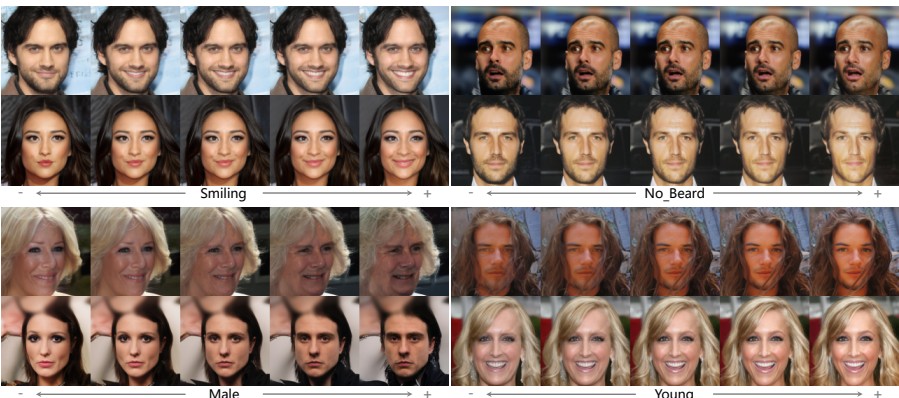

Figure 8: Attribute manipulation examples generated by "CelebA-HQ128-52M-z512-25M". For each example, we manipulate the input image (middle) by moving its semantic latent code along the direction of corresponding attribute found by trained linear classifiers with different scales.

$\boldsymbol{G}_\psi\big(\boldsymbol{x}_t, Lerp(\boldsymbol{z}^1, \boldsymbol{z}^2; \lambda), t\big) \approx Lerp\big(\boldsymbol{G}_\psi(\boldsymbol{x}_t, \boldsymbol{z}^1, t), \boldsymbol{G}_\psi(\boldsymbol{x}_t, \boldsymbol{z}^2, t); \lambda\big)$, so that $\boldsymbol{G}_\psi(\boldsymbol{x}_t, \boldsymbol{z}, t)$ can be seen as a function of $\boldsymbol{z}$ analogous to a linear map. The linearity guarantees a meaningful semantic latent space that represents the semantic spatial change of image by a linear change of latent code.

### 4.5 Attribute Manipulation

We can further explore the learned semantic latent space in a supervised way. To illustrate this, we train a model of "CelebA-HQ128-52M-z512-25M" and conduct attribute manipulation experiments by utilizing the attribute annotations of CelebA-HQ dataset. Specifically, we first encode an image to its semantic latent code, then move it along the learned direction and finally decode it to the manipulated image. Similar to Diff-AE, we train a linear classifier to separate the semantic latent codes of the images with different attribute labels and use the normal vector of separating hyperplane (i.e. the weight of linear classifier) as the direction vector. We present some attribute manipulation examples in Figure 8. As we can see, PDAE succeeds in manipulating images by moving their semantic latent codes along the direction of desired attribute with different scales. Like Diff-AE, PDAE can change attribute-relevant features while keeping other irrelevant details almost stationary if using the inferred $\boldsymbol{x}_T$ of input image.

### 4.6 Truncation-like Effect

According to [8, 15], we can obtain a truncation-like effect in DPMs by scaling the strength of classifier guidance. We have assumed that $\boldsymbol{G}_\psi(\boldsymbol{x}_t, \boldsymbol{z}, t)$ trained by filling the posterior mean gap simulates the gradient of some implicit classifier, and it can actually work as desired. In theory, it can

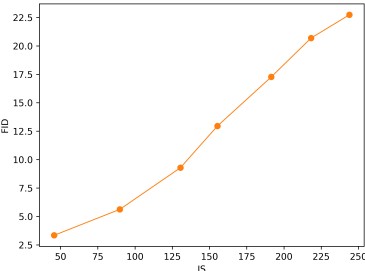

Figure 9: The truncation-like effect for "ImageNet64-77M-y-38M" by scaling $\boldsymbol{G}_\psi(\boldsymbol{x}_t, \boldsymbol{y}, t)$ with 0.0, 0.5, 1.0, 1.5, 2.0, 2.5, 3.0 respectively.

| Dataset | Model | FID | | | |
|---|---|---|---|---|---|
| | | T=10 | T=20 | T=50 | T=100 |
| FFHQ | DDIM | 31.87 | 20.53 | 15.82 | 11.95 |
| | Diff-AE | 21.95 | 18.10 | 13.14 | 10.55 |
| | PDAE | **20.16** | **17.18** | **12.81** | **10.31** |
| Horse [55] | DDIM | 25.24 | 14.41 | 7.98 | 5.93 |
| | Diff-AE | 12.66 | 9.21 | 7.12 | 5.27 |
| | PDAE | **11.94** | **8.51** | **6.83** | **5.09** |
| Bedroom [55] | DDIM | 14.07 | 9.29 | 7.31 | 5.88 |
| | Diff-AE | 10.79 | 8.42 | 6.49 | **5.32** |
| | PDAE | **10.05** | **7.89** | **6.33** | 5.47 |
| CelebA | DDIM | 18.89 | 13.82 | 8.48 | 5.94 |
| | Diff-AE | 12.92 | 10.18 | **7.05** | 5.30 |
| | PDAE | **11.84** | **9.65** | 7.23 | **5.19** |

Table 2: FID scores for unconditional sampling.

also be applied in truncation-like effect. To illustrate this, we directly incorporate the class label into $\boldsymbol{G}_\psi(\boldsymbol{x}_t, \boldsymbol{y}, t)$ and train it to fill the gap. Specifically, we train a model of "ImageNet64-77M-y-38M" and use DDIM sampling method with 100 steps to generate 50k samples, guided by the predicted mean shift with different scales for a truncation-like effect. Figure 9 shows the sample quality effects of sweeping over the scale. As we can see, it achieves the truncation-like effect similar to that of classifier-guided sampling method, which helps us to build connections between filling the posterior mean gap and classifier-guided sampling method. The gradient estimator trained by filling the posterior mean gap is an alternative to the noisy classifier.

### 4.7 Few-shot Conditional Generation

Following D2C [42], we train a model of "CelebA64-72M-z512-38M" on CelebA [20] and aim to achieve conditional sampling given a small number of labeled images. To achieve this, we train a latent DPM $p_\omega(\boldsymbol{z}_{t-1}|\boldsymbol{z}_t)$ on semantic latent space and a latent classifier $p_\eta(y|\boldsymbol{z})$ using given labeled images. For binary scenario, the images are labeled by a binary class (100 samples, 50 for each class).

For PU scenario, the images are either labeled positive or unlabeled (100 positively labeled and 10k unlabeled samples). Then we sample $\boldsymbol{z}$ from $p_\omega(\boldsymbol{z}_{t-1}|\boldsymbol{z}_t)$ and accept it with the probability of $p_\eta(y|\boldsymbol{z})$. We use the accepted $\boldsymbol{z}$ to generate 5k samples for every class and compute the FID score between these samples and all images belonging to corresponding class in dataset. We compare PDAE with Diff-AE and D2C. We also use the naive approach that computes the FID score between the training images and the corresponding subset of images in dataset. Table 3 shows that PDAE achieves better FID scores than Diff-AE and D2C.

Table 3: FID scores for few-shot conditional generation using "CelebA64-72M-z512-38M".

| Scenario | Classes | PDAE | Diff-AE [36] | D2C [42] | Naive |
|---|---|---|---|---|---|
| Binary | Male | **11.21** | 11.52 | 13.44 | 25.70 |
| | Female | **6.81** | 7.29 | 9.51 | 14.16 |
| | Blond | 16.96 | **16.10** | 17.61 | 24.78 |
| | Non-Blond | **8.13** | 8.48 | 8.94 | 1.12 |
| PU | Male | **9.41** | 9.54 | 16.39 | 25.70 |
| | Female | **8.97** | 9.21 | 12.21 | 14.16 |
| | Blond | **6.34** | 7.01 | 10.09 | 24.78 |
| | Non-Blond | **7.17** | 7.91 | 9.09 | 1.12 |

### 4.8 Improved Unconditional Sampling

As shown in Section 4.2, under the help of $\boldsymbol{z}$, PDAE can generate plausible images in only one step. If we can get $\boldsymbol{z}$ in advance, PDAE can achieve better sample quality than pre-trained DPMs in the same number of sampling steps. Similar to Diff-AE, we train a latent DPM on semantic latent space and sample $\boldsymbol{z}$ from it to improve the unconditional sampling of pre-trained DPMs.

Unlike Diff-AE that must take $\boldsymbol{z}$ as input for sampling, PDAE uses an independent gradient estimator as a corrector of the pre-trained DPM for sampling. We find that only using pre-trained DPMs in the last few sampling steps can achieve better sample quality, which may be because that the gradient estimator is sensitive to $\boldsymbol{z}$ in the last few sampling steps and the stochasticity of sampled

$z$ will lead to out-of-domain samples. Asyrp [27] also finds similar phenomenon. Empirically, we carry out this strategy in the last 30% sampling steps. We evaluate unconditional sampling result on "FFHQ128-130M-z512-64M", "Horse128-130M-z512-64M", "Bedroom128-120M-z512-70M" and "CelebA64-72M-z512-38M" using DDIM sampling method with different steps. For each dataset, we calculate the FID scores between 50k generated samples and 50k real images randomly selected from dataset. Table 2 shows that PDAE significantly improves the sample quality of pre-trained DPMs and outperforms Diff-AE. Note that PDAE can be applied for any pre-trained DPMs as an auxiliary booster to improve their sample quality.

## 5 Related Work

Our work is based on an emerging latent variable generative model known as Diffusion Probabilistic Models (DPMs) [43, 14], which are now popular for their stable training process and competitive sample quality. Numerous studies [34, 24, 8, 15, 44, 19, 46, 30] and applications [5, 26, 18, 32, 57, 6, 29, 41, 3, 16, 17] have further significantly improved and expanded DPMs.

Unsupervised representation learning via generative modeling is a popular topic in computer vision. Latent variable generative models, such as GANs [13], VAEs [25, 39], and DPMs, are a natural candidate for this, since they inherently involve a latent representation of the data they generate. For GANs, due to its lack of inference functionality, one have to extract the representations for any given real samples by an extra technique called GAN Inversion [54], which invert samples back into the latent space of trained GANs. Existing inversion methods [58, 35, 4, 1, 2, 51] either have limited reconstruction quality or need significantly higher computational cost. VAEs explicitly learn representations for samples, but still face representation-generation trade-off challenges [49, 42]. VQ-VAE [49, 37] and D2C [42] overcome these problems by modeling latent variables post-hoc in different ways. DPMs also yield latent variables through the forward process. However, these latent variables lack high-level semantic information because they are just a sequence of spatially corrupted images. In light of this, diffusion autoencoders (Diff-AE) [36] explore DPMs for representation learning via autoencoding. Specifically, they jointly train an encoder for discovering meaningful representations from images and a conditional DPM as the decoder for image reconstruction by treating the representations as input conditions. Diff-AE is competitive with the state-of-the-art model on image reconstruction and capable of various downstream tasks. Compared with Diff-AE, PDAE leverages existing pre-trained DPMs for representation learning also via autoencoding, but with better training efficiency and performance.

A concurrent work with the similar idea is the textual inversion of pre-trained text-to-image DPMs [12]. Specifically, given only 3-5 images of a user-provided concept, like an object or a style, they learn to represent it through new "words" in the embedding space of the frozen text-to-image DPMs. These learned "words" can be further composed into natural language sentences, guiding personalized creation in an intuitive way. From the perspective of posterior mean gap, for the given new concept, textual inversion optimizes its corresponding new "words" embedding vector to find a best textual condition $(c)$, so that which can be fed into pre-trained text-to-image DPMs $(\epsilon_\theta(x_t, c, t))$ to fill as much gap $(\epsilon - \epsilon_\theta(x_t, \emptyset, t))$ as possible.

## 6 Conclusion

In conclusion, we present a general method called PDAE that leverages pre-trained DPMs for representation learning via autoencoding and achieves better training efficiency and performance than Diff-AE. Our key idea is based on the concept of posterior mean gap and its connections with classifier-guided sampling method. A concurrent work, textual inversion of pre-trained text-to-image DPMs, can also be explained from this perspective. We think the idea can be further explored to extract knowledges from pre-trained DPMs, such as interpretable direction discovery [51], and we leave it as future work.

## Acknowledgments and Disclosure of Funding

This work was supported in part by the National Natural Science Foundation of China (Grant No. 62072397 and No.61836002), Zhejiang Natural Science Foundation (LR19F020006) and Yiwise.

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
