# Unsupervised Representation Learning from Pre-trained Diffusion Probabilistic Models Appendix

## A Algorithm

Algorithm 1 shows the training procedure of PDAE. Algorithm 2 3 show the DDPM and DDIM sampling procedure of PDAE, respectively.

---
**Algorithm 1:** Training

---
**Prepare:** dataset distribution $p_{data}(\boldsymbol{x}_0)$, pre-trained DPM $(\boldsymbol{\epsilon}_\theta, \boldsymbol{\Sigma}_\theta)$.
**Initialize:** encoder $\boldsymbol{E}_\varphi$, gradient-estimator $\boldsymbol{G}_\psi$.
**Run:**
**repeat**

> $\boldsymbol{x}_0 \sim p_{data}(\boldsymbol{x}_0)$
> $t \sim \text{Uniform}(1, 2, \cdots, T)$
> $\boldsymbol{\epsilon} \sim \mathcal{N}(\boldsymbol{0}, \mathbf{I})$
> $\boldsymbol{x}_t = \sqrt{\bar{\alpha}_t}\boldsymbol{x}_0 + \sqrt{1 - \bar{\alpha}_t}\boldsymbol{\epsilon}$
> Update $\varphi$ and $\psi$ by taking gradient descent step on
> $\nabla_{\varphi,\psi}\, \lambda_t \big\| \epsilon - \boldsymbol{\epsilon}_\theta(\boldsymbol{x}_t, t) + \frac{\sqrt{\alpha_t}\sqrt{1-\bar{\alpha}_t}}{\beta_t} \cdot \boldsymbol{\Sigma}_\theta(\boldsymbol{x}_t, t) \cdot \boldsymbol{G}_\psi(\boldsymbol{x}_t, \boldsymbol{E}_\varphi(\boldsymbol{x}_0), t) \big\|^2$

**until** *converged*;

---

Usually we set $\boldsymbol{\Sigma}_\theta = \sigma_t^2 \mathbf{I} = \frac{1-\bar{\alpha}_{t-1}}{1-\bar{\alpha}_t}\beta_t\mathbf{I}$ to untrained time-dependent constants.

---
**Algorithm 2:** DDPM Sampling(Autoencoding)

---
**Prepare:** pre-trained DPM $\boldsymbol{\epsilon}_\theta$, trained encoder $\boldsymbol{E}_\varphi$, trained gradient estimator $\boldsymbol{G}_\psi$,
**Input:** sample $\boldsymbol{x}$
**Run:**
$\boldsymbol{z} = \boldsymbol{E}_\varphi(\boldsymbol{x})$
$\boldsymbol{x}_T \sim \mathcal{N}(\boldsymbol{0}, \mathbf{I})$
**for** $t = T$ **to** $1$ **do**

> $\boldsymbol{\epsilon} \sim \mathcal{N}(\boldsymbol{0}, \mathbf{I})$ if $t \geq 2$, else $\boldsymbol{\epsilon} = \boldsymbol{0}$
> $\boldsymbol{x}_{t-1} = \frac{1}{\sqrt{\alpha_t}}\left[ \boldsymbol{x}_t - \frac{\beta_t}{\sqrt{1-\bar{\alpha}_t}}\boldsymbol{\epsilon}_\theta(\boldsymbol{x}_t, t) \right] + \sigma_t^2 \boldsymbol{G}_\psi(\boldsymbol{x}_t, \boldsymbol{z}, t) + \sigma_t\boldsymbol{\epsilon}$

**return** $\boldsymbol{x}_0$

---

---
**Algorithm 3:** DDIM Sampling(Autoencoding)

---
**Prepare:** pre-trained DPM $\boldsymbol{\epsilon}_\theta$, trained encoder $\boldsymbol{E}_\varphi$, trained gradient estimator $\boldsymbol{G}_\psi$,
**Input:** sample $\boldsymbol{x}$, sampling sequence $\{t_i\}_{i=1}^S$ where $t_1 = 0$ and $t_S = T$
**Run:**
$\boldsymbol{z} = \boldsymbol{E}_\varphi(\boldsymbol{x})$
$\boldsymbol{x}_T \sim \mathcal{N}(\boldsymbol{0}, \mathbf{I})$ or use inferred $\boldsymbol{x}_T$
**for** $i = S$ **to** $2$ **do**

> $\hat{\boldsymbol{\epsilon}}_\theta(\boldsymbol{x}_{t_i}, t_i) = \boldsymbol{\epsilon}_\theta(\boldsymbol{x}_{t_i}, t_i) - \sqrt{1 - \bar{\alpha}_{t_i}} \cdot \boldsymbol{G}_\psi(\boldsymbol{x}_{t_i}, \boldsymbol{z}, t_i)$
> $\boldsymbol{x}_{t_{i-1}} = \sqrt{\bar{\alpha}_{t_{i-1}}}\left( \frac{\boldsymbol{x}_t - \sqrt{1-\bar{\alpha}_{t_i}}\hat{\boldsymbol{\epsilon}}_\theta(\boldsymbol{x}_{t_i}, t_i)}{\sqrt{\bar{\alpha}_{t_i}}} \right) + \sqrt{1 - \bar{\alpha}_{t_{i-1}}} \cdot \hat{\boldsymbol{\epsilon}}_\theta(\boldsymbol{x}_{t_i}, t_i)$

**return** $\boldsymbol{x}_0$

---

36th Conference on Neural Information Processing Systems (NeurIPS 2022).

# B    Implementation Details

## B.1    Network Architecture

Table 1 shows the network architecture of pre-trained DPMs we use. $\boldsymbol{G}_\psi$ is completely determined by pre-trained DPMs. For $\boldsymbol{E}_\varphi$, we use stacked GroupNorm-SiLU-Conv layers to convert input images into $256 \times 4 \times 4$ feature maps and a linear layer to map it into $\boldsymbol{z}$. A self-attention block is employed at $16 \times 16$ resolution.

Table 1: Network architecture of pre-trained DPMs based on ADM [1] in guided-diffusion. We use pre-trained DPMs provided by Diff-AE [2] in official Diff-AE implementation.

| Parameter | CelebA 64 | CelebA-HQ 128 | FFHQ 128 | Horse 128 | Bedroom 128 |
|---|---|---|---|---|---|
| Base channels | 64 | 128 | 128 | 128 | 128 |
| Channel multipliers | [1,2,4,8] | [1,1,2,3,4] | [1,1,2,3,4] | [1,1,2,3,4] | [1,1,2,3,4] |
| Attention resolutions | | | [16] | | |
| Attention heads num | 4 | 1 | 1 | 1 | 1 |
| Dropout | | | 0.1 | | |
| Images trained | 72M | 52M | 130M | 130M | 120M |
| $\beta$ scheduler | | | Linear | | |
| Training $T$ | | | 1000 | | |
| Diffusion loss | | | MSE with noise prediction $\epsilon$ | | |

Table 2: Network architecture of latent DPMs.

| Parameter | CelebA 64 | FFHQ 128 | Horse 128 | Bedroom 128 |
|---|---|---|---|---|
| MLP layers ($N$) | 10 | 10 | 20 | 20 |
| MLP hidden size | | 2048 | | |
| Batch size | | 512 | | |
| Optimizer | | Adam (no weight decay) | | |
| Learning rate | | 1e-4 | | |
| EMA rate | | 0.9999/batch | | |
| $\beta$ scheduler | | Constant 0.008 | | |
| Training T | | 1000 | | |
| Diffusion loss | | L1 loss with noise prediction $\epsilon$ | | |

For fair comparison, we follow Diff-AE [2] to use deep MLPs as the denoising network of latent DPMs. Table 2 shows the network architecture. Specifically, we calculate $\boldsymbol{z} = \boldsymbol{E}_\varphi(\boldsymbol{x}_0)$ for all $\boldsymbol{x}_0$ from dataset and normalize them to zero mean and unit variance. Then we learn the latent DPMs $p_\omega(\boldsymbol{z}_{t-1}|\boldsymbol{z}_t)$ by optimizing:

$$\mathcal{L}(\omega) = \mathbb{E}_{\boldsymbol{z},t,\epsilon}\big[\|\epsilon - \boldsymbol{\epsilon}_\omega(\boldsymbol{z}_t, t)\|\big], \tag{1}$$

where $\epsilon \sim \mathcal{N}(\boldsymbol{0}, \mathbf{I})$ and $\boldsymbol{z}_t = \sqrt{\bar{\alpha}_t}\boldsymbol{z} + \sqrt{1 - \bar{\alpha}_t}\epsilon$. The sampled $\boldsymbol{z}$ will be denormalized for use.

## B.2    Experimental Details

During the training of PDAE, we set batch size as 128 for all datasets. We always set learning rate as $1e - 4$ and use $512\text{-}d$ $\boldsymbol{z}$. We use EMA on all model parameters with a decay factor of 0.9999.

For attribute manipulation, we train a linear classifier to separate the normalized semantic latent codes of the images with different attribute labels. During manipulation, we first normalize $\boldsymbol{z} = \boldsymbol{E}_\varphi(\boldsymbol{x}_0)$ to zero mean and unit variance, then move it towards the normal vector of separating hyperplane (i.e. the weight of linear classifier) with different scales, finally denormalize it for sampling.

For few-shot conditional generation, we follow [3] to train PU classifier by oversampling positively labeled samples to balance the batch samples. During conditional generation of class $y$, for a sampled $\boldsymbol{z}$, we reject it when $p_\eta(y|\boldsymbol{z}) < 0.5$ and accept it with the probability of $p_\eta(y|\boldsymbol{z})$ when $p_\eta(y|\boldsymbol{z}) \geq 0.5$.

# C  Additional Samples

## C.1  Learned Mean Shift Fills Posterior Mean Gap

Figure 1 2 3 show the predicted $\hat{x}_0$ by denoising $x_t = \sqrt{\bar{\alpha}_t}x_0 + \sqrt{1-\bar{\alpha}_t}\epsilon$ for only one step using different models. Figure 4 shows the calculated average posterior mean gap for $\|\tilde{\mu}_t(x_t, x_0) - \mu_\theta(x_t, t)\|^2$ and $\|\tilde{\mu}_t(x_t, x_0) - (\mu_\theta(x_t, t) + \Sigma_\theta(x_t, t) \cdot G_\psi(x_t, E_\varphi(x_0), t))\|^2$. As we can see, PDAE can predict the mean shift that indeed fills the posterior mean gap.

## C.2  Autoencoding Reconstruction

Figure 5 6 7 show some autoencoding reconstruction examples using different models. As we can see, the deterministic method can almost reconstruct the input images even with only 100 steps and both stochastic methods can generate samples with similar contents to the input except some minor details, such as sheet pattern and wrinkle for LSUN-Bedroom; horse eye, spot and mane for LSUN-Horse.

## C.3  Interpolation of Semantic Latent Codes and Trajectories

Figure 8 9 10 show some examples of two kinds of interpolation methods using different models. Due to complex scenes for images of LSUN-Bedroom and LSUN-Horse, we manually select some spatially-similar image pairs for interpolation. As we can see, both methods generate similar samples that smoothly transition from one endpoint to the other.

## C.4  Attribute Manipulation

Figure 11 shows some attribute manipulation examples. As we can see, PDAE succeeds in manipulating images by moving their semantic latent codes along the direction of desired attribute with different scales. Like Diff-AE, PDAE can change attribute-relevant features while keeping other irrelevant details almost stationary if using the inferred $x_T$ of input image.

## C.5  Few-shot Conditional Generation

We present some samples for 4 PU scenarios of few-shot conditional generation in Figure 12. As we can see, PDAE can generate samples belonging to specified class for different few-shot scenarios, which shows that our semantic latent codes are easy to classify even with a very small number of labeled samples.

## C.6  Visualization of Mean Shift

We visualize some examples of $G_\psi(x_t, E_\varphi(x_0), t)$ in Figure 13. As we can see, the gradient estimator learns a mean shift direction towards $x_0$ for each $x_t$.

# D  Limitations and Potential Negative Societal Impacts

Although better training efficiency, PDAE has a slower inference speed than Diff-AE due to an extra gradient estimator, which also needs more memory and storage space.

Slow generation speed is a common problem for DPM-based works. Although many studies have been able to achieve decent performance with few reverse steps, they still lag behind VAEs and GANs, which only need a single network pass. Furthermore, almost perfect PDAE reconstruction needs hundreds of extra forward steps to infer the stochastic latent code.

Moreover, we have found that the weighting scheme of diffusion loss is indispensable to PDAE, but we haven't explored its mechanism, which may help to further improve the efficiency and performance of PDAE. We leave empirical and theoretical investigations of this aspect as future work.

Potential negative impacts of our work mainly involve deepfakes, which leverage powerful generative techniques from machine learning and artificial intelligence to create synthetic media, which may be used for hoaxes, fraud, bullying or revenge. Although some synthetic samples are hard to distinguish, researchers have developed algorithms similar to the ones used to build the deepfake to detect

them with high accuracy. Some other techniques such as blockchain and digitally signing can help platforms to verify the source of the media.

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

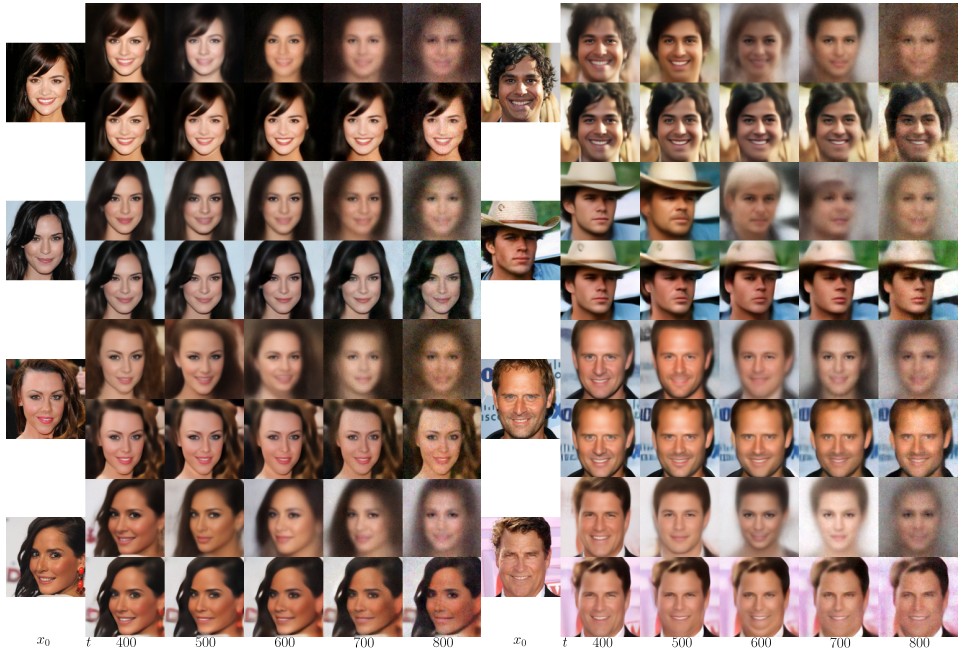

Figure 1: Predicted $\hat{x}_0$ by denoising $x_t$ for only one step using "CelebA-HQ128-52M-z512-25M". The first row use pre-trained DPM and the second row use PDAE.

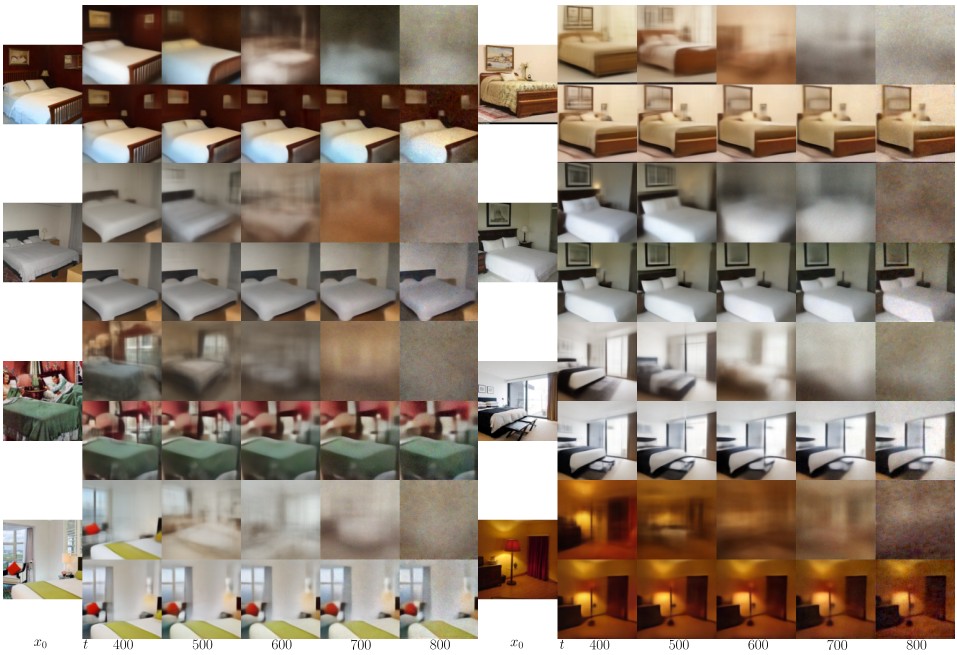

Figure 2: Predicted $\hat{x}_0$ by denoising $x_t$ for only one step using "Bedroom128-120M-z512-70M". The first row use pre-trained DPM and the second row use PDAE.

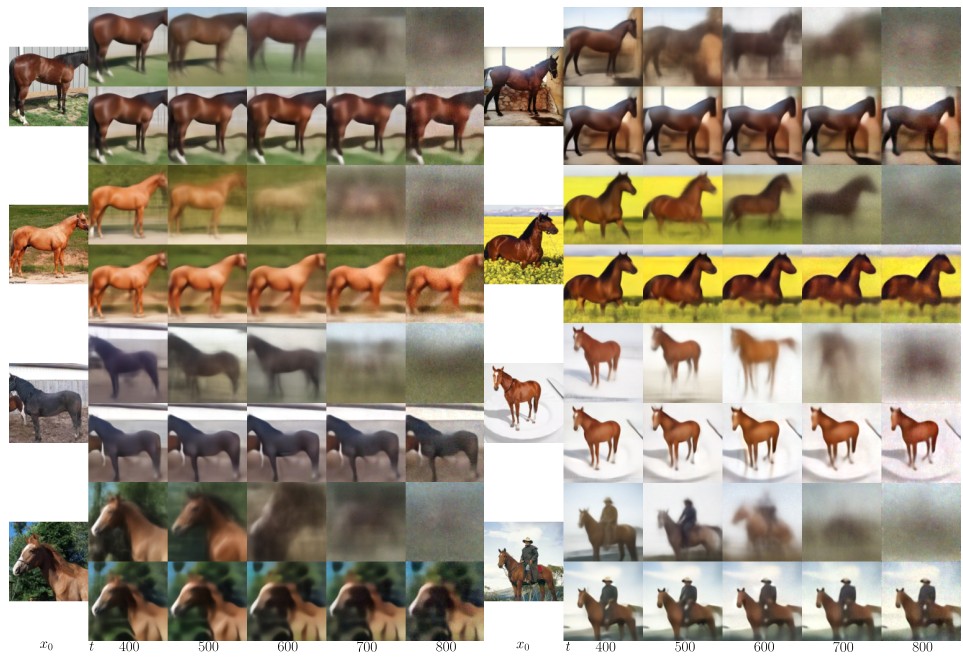

Figure 3: Predicted $\hat{x}_0$ by denoising $x_t$ for only one step using "Horse128-130M-z512-64M". The first row use pre-trained DPM and the second row use PDAE.

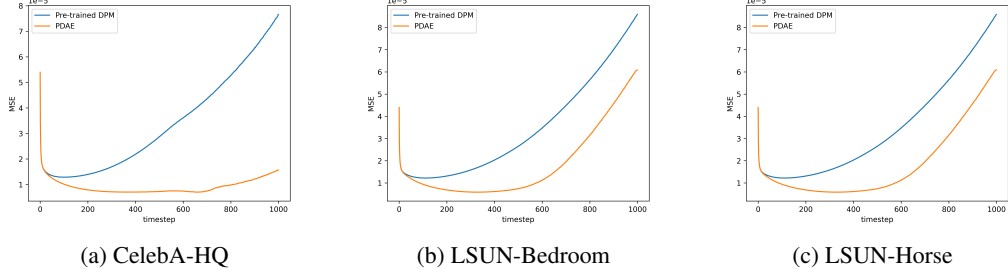

(a) CelebA-HQ      (b) LSUN-Bedroom      (c) LSUN-Horse

Figure 4: Average posterior mean gap (calculated on 1000 randomly selected images).

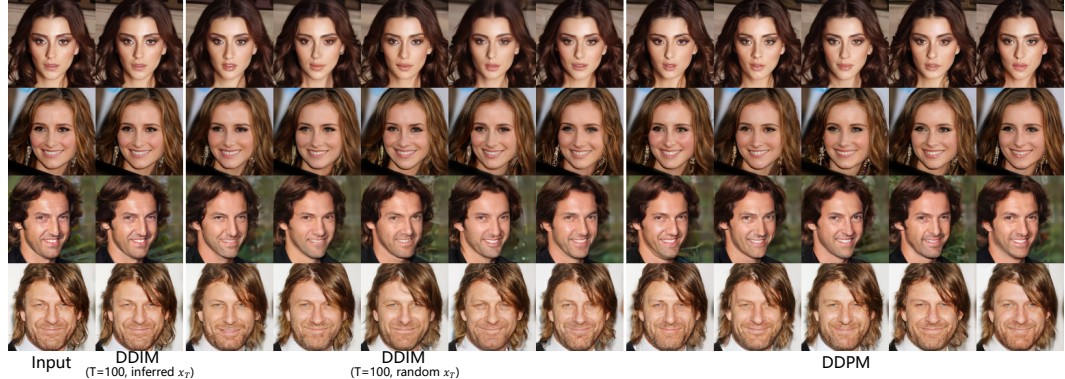

Input    DDIM            DDIM                          DDPM
(T=100, inferred $x_T$)     (T=100, random $x_T$)

Figure 5: Autoencoding reconstruction examples generated by "CelebA-HQ128-52M-z512-25M" with different sampling methods. Each row corresponds to an exmaple.

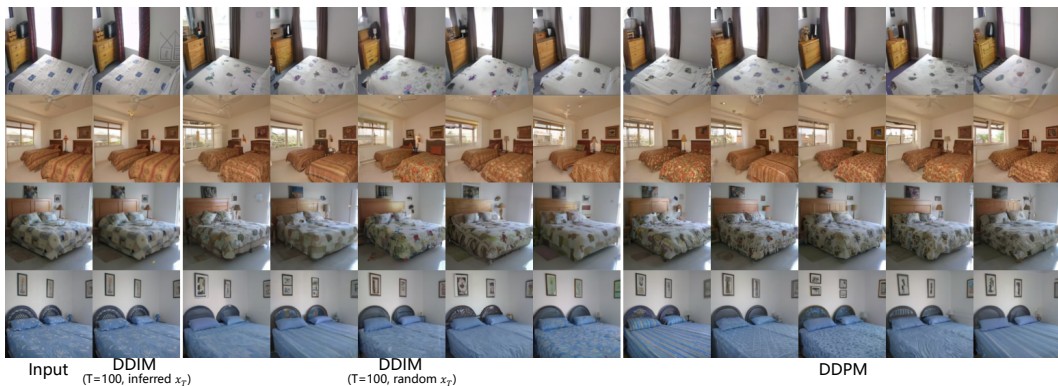

Input    DDIM            DDIM                          DDPM
(T=100, inferred $x_T$)     (T=100, random $x_T$)

Figure 6: Autoencoding reconstruction examples generated by "Bedroom128-120M-z512-70M" with different sampling methods. Each row corresponds to an exmaple.

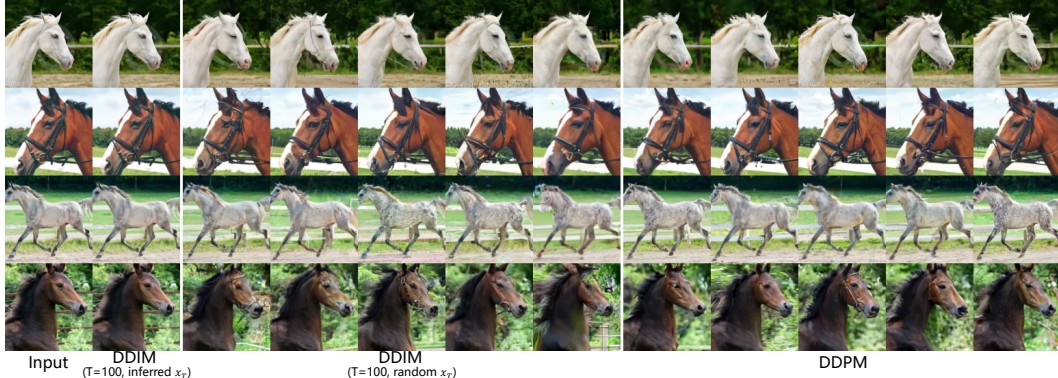

Input    DDIM            DDIM                          DDPM
(T=100, inferred $x_T$)     (T=100, random $x_T$)

Figure 7: Autoencoding reconstruction examples generated by "Horse128-130M-z512-64M" with different sampling methods. Each row corresponds to an exmaple.

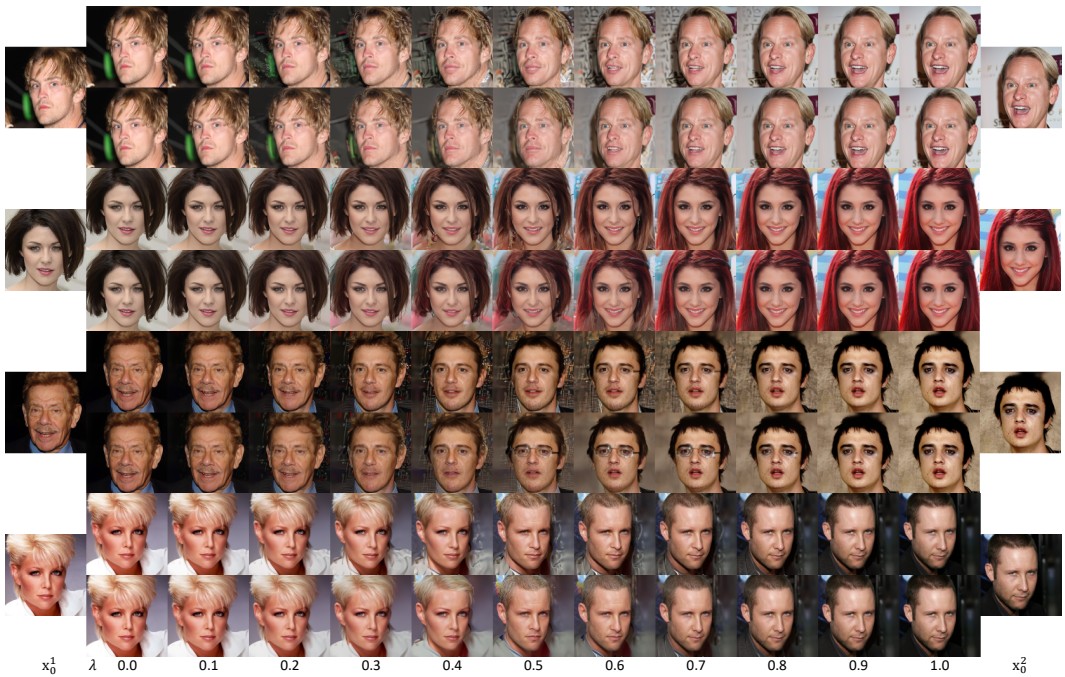

Figure 8: Interpolation examples generated by "CelebA-HQ128-52M-z512-25M". For each example, the first row use the guidance of $\boldsymbol{G}_\psi\big(\boldsymbol{x}_t, Lerp(\boldsymbol{z}^1, \boldsymbol{z}^2; \lambda), t\big)$ and the second row use the guidance of $Lerp\big(\boldsymbol{G}_\psi(\boldsymbol{x}_t, \boldsymbol{z}^1, t), \boldsymbol{G}_\psi(\boldsymbol{x}_t, \boldsymbol{z}^2, t); \lambda\big)$.

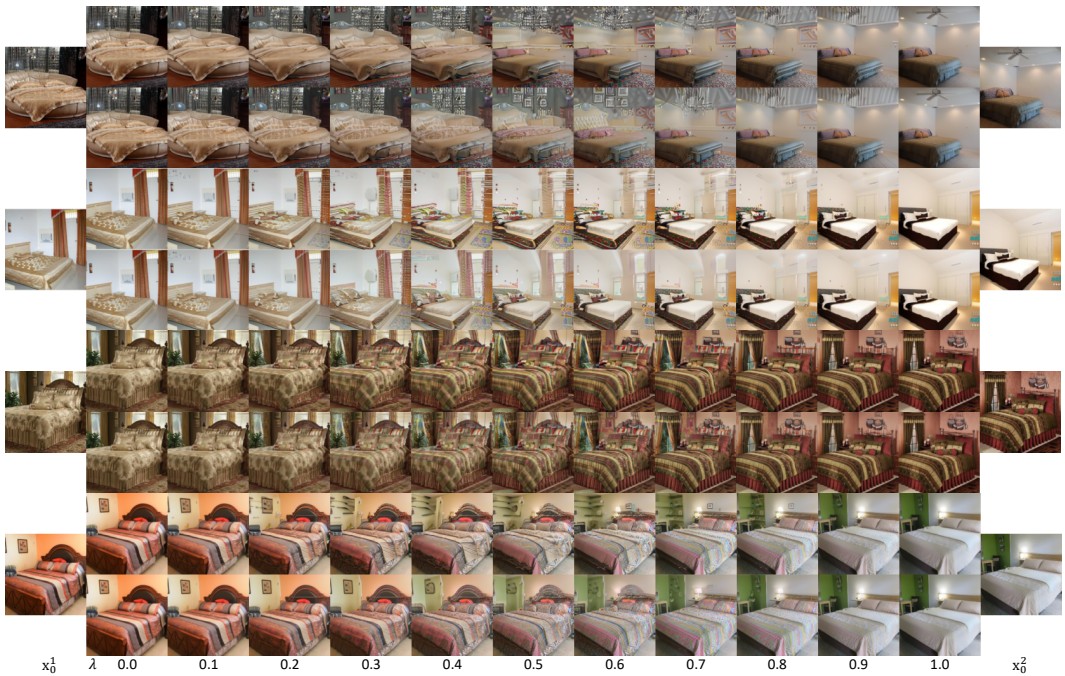

Figure 9: Interpolation examples generated by "Bedroom128-120M-z512-70M". For each example, the first row use the guidance of $\boldsymbol{G}_\psi\big(\boldsymbol{x}_t, Lerp(\boldsymbol{z}^1, \boldsymbol{z}^2; \lambda), t\big)$ and the second row use the guidance of $Lerp\big(\boldsymbol{G}_\psi(\boldsymbol{x}_t, \boldsymbol{z}^1, t), \boldsymbol{G}_\psi(\boldsymbol{x}_t, \boldsymbol{z}^2, t); \lambda\big)$.

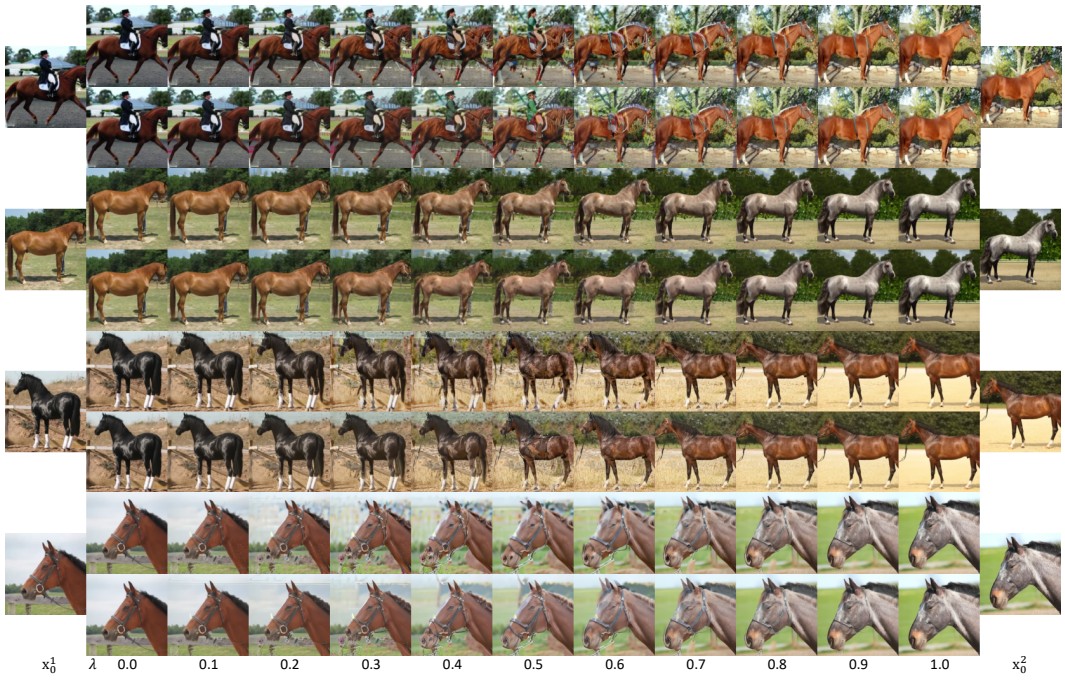

| $x_0^1$ | $\lambda$ | 0.0 | 0.1 | 0.2 | 0.3 | 0.4 | 0.5 | 0.6 | 0.7 | 0.8 | 0.9 | 1.0 | $x_0^2$ |

Figure 10: Interpolation examples generated by "Horse128-130M-z512-64M". For each example, the first row use the guidance of $\boldsymbol{G}_\psi\big(\boldsymbol{x}_t, Lerp(\boldsymbol{z}^1, \boldsymbol{z}^2; \lambda), t\big)$ and the second row use the guidance of $Lerp\big(\boldsymbol{G}_\psi(\boldsymbol{x}_t, \boldsymbol{z}^1, t), \boldsymbol{G}_\psi(\boldsymbol{x}_t, \boldsymbol{z}^2, t); \lambda\big)$.

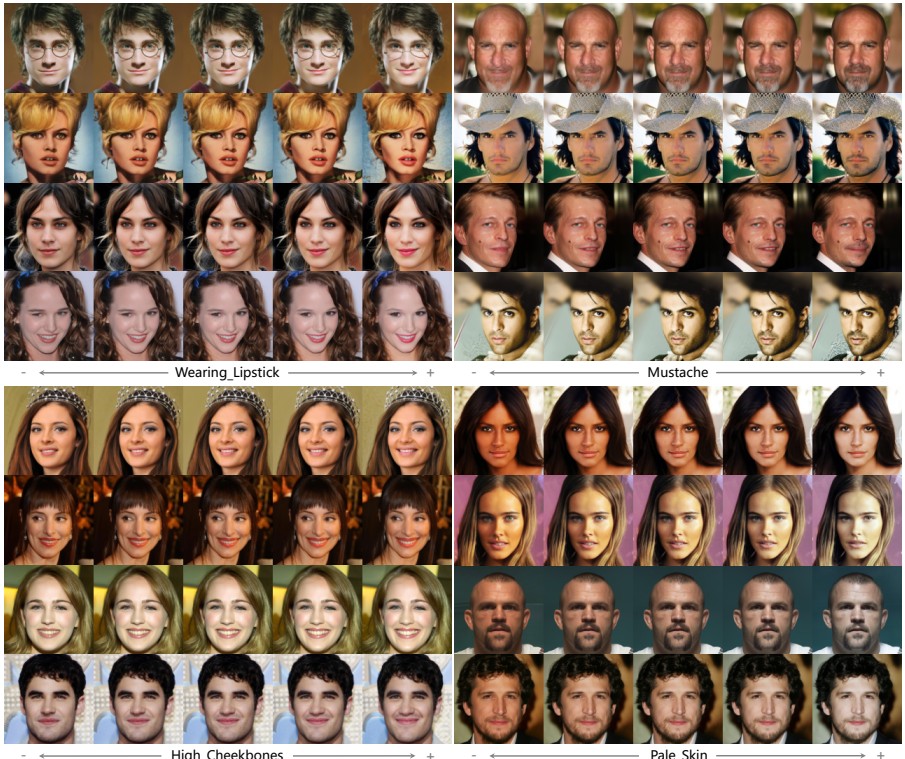

Figure 11: Attribute manipulation examples generated by "CelebA-HQ128-52M-z512-25M". For each example, we manipulate the input image (middle) by moving its semantic latent code along the direction of corresponding attribute found by trained linear classifiers with different scales.

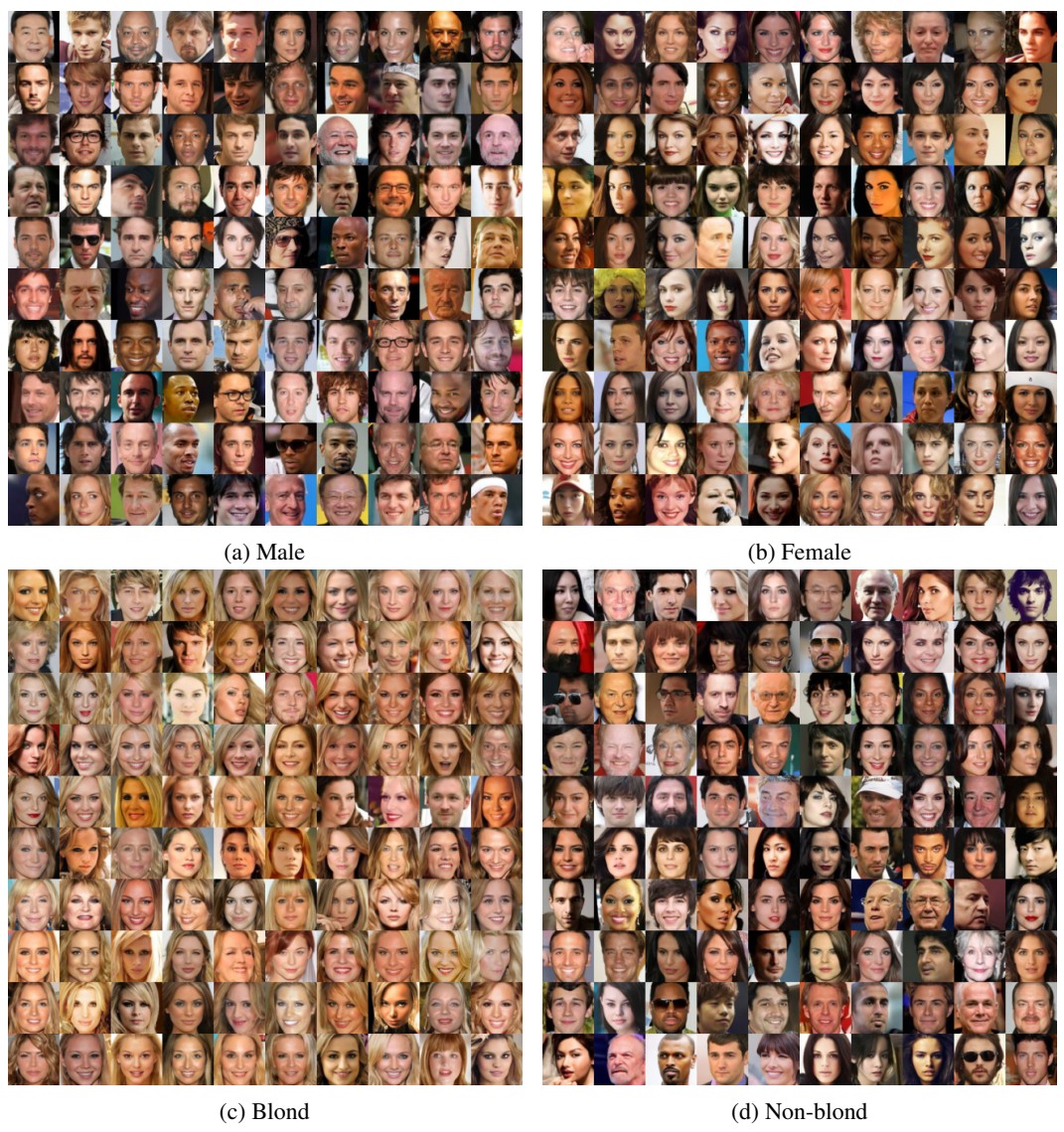

(a) Male             (b) Female

(c) Blond             (d) Non-blond

Figure 12: Samples for 4 PU scenarios of few-shot conditional generation using "CelebA64-72M-z512-38M".

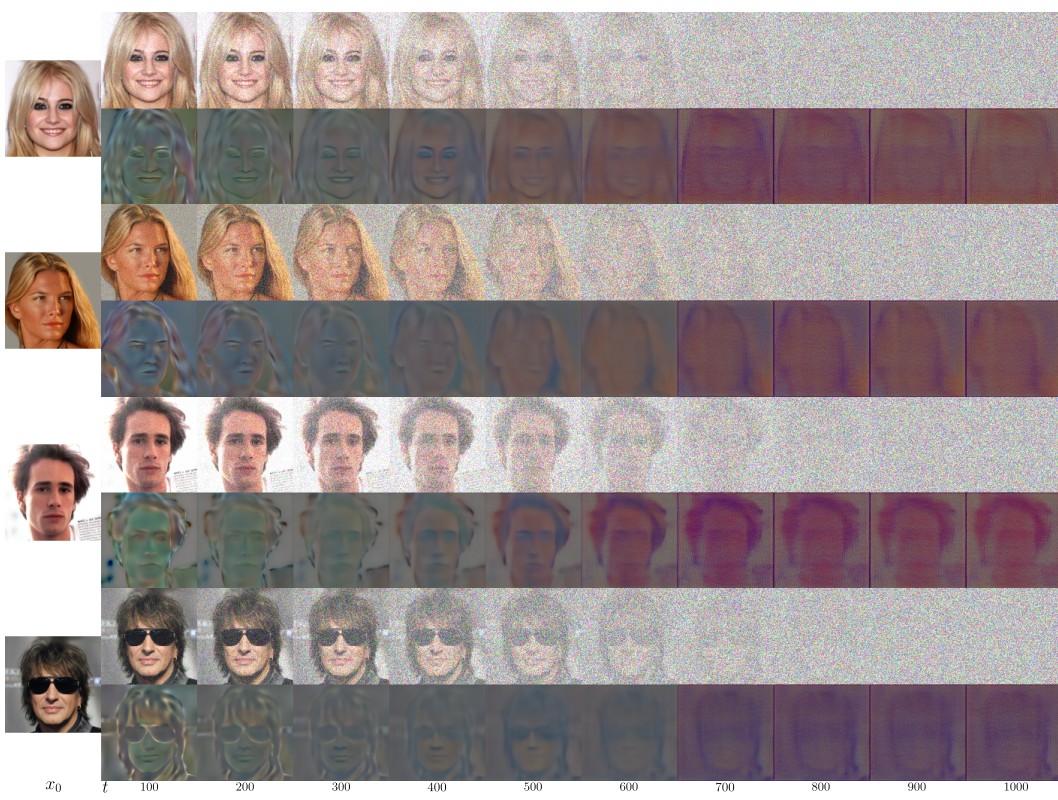

Figure 13: Visualization of mean shift generated by "CelebA-HQ128-52M-z512-25M". For each example, the first row shows $\boldsymbol{x}_t$ and the second row shows $\boldsymbol{G}_\psi(\boldsymbol{x}_t, \boldsymbol{E}_\varphi(\boldsymbol{x}_0), t)$.