# OpenReview forum: "Unsupervised Representation Learning from Pre-trained Diffusion Probabilistic Models"
_NeurIPS.cc/2022/Conference — NeurIPS 2022 Accept_

### Official Review · Reviewer_4H5f · 2022-07-02

**Rating:** 6
**Confidence:** 3
**Soundness:** 3 good
**Presentation:** 3 good
**Contribution:** 3 good

**Summary:**

The authors propose an unsupervised method for data representation and reconstruction, utilizing a pre-trained unconditional diffusion probabilistic model (DMP) in order to avoid the long training time of such models.
The method incorporates a gradient estimator, that learns the posterior mean shift, which is then used as a condition in order to guide the sampling process towards the image reconstruction. The authors also propose a new weighting scheme for training the objective function. Learning the gradient estimator encourages the encoder to learn meaningful data representation, and the sampling process to produce better data reconstruction.
The authors provide quantitative and qualitative evaluations showing improvements in representation learning, image reconstruction, and image sampling.

**Questions:**

- Line 16: "and train it like a normal DMP": this causes some confusion, as the DMP part of the model is fixed. Also, the objective function aims to learn image reconstruction rather than image generation.

- It would be nice to see the results of the experiment shown in section 3.4, but when using the gradient-estimator as classifier-guided sampling, instead of a separate classifier, as done on MNIST.

- In the experiment section, the prediction of x_0 is done from x_t in only one step. What does this "one-step" mean? Isn't the whole reverse process required in order to retrieve x_0?

- It would be great to have an experiment that supports the claim that this method requires less training time than its competitors.

**Limitations:**

The author did not address their method's limitations.
The described work has no potential negative societal impact.

**Strengths And Weaknesses:**

Strength:

- The problem is known to be valid and challenging, and the level of novelty is reasonable.

- The method, idea, and motivation are well explained in the paper. The authors provided a good intuition about their proposed network architecture.

- The authors well-explained their weighting method by describing their insights from an experiment that shows differences in the result with and without the classifier-guided sampling

Weaknesses:

- There is no algorithm that explains how the sampling step works in the author's method (how does the sampling-guided method is employed when using the model described in the paper).

- I suggest showing Figures 1,2 in separate locations (figure 2 should be closer to the text that describes it, along with figure 3).

- According to the quantitative results, the improvement of data representation is not significant. However, there is an improvement in the rest of the experiments, such as in unconditional sampling.

- Overall proofreading is required.

---

> ### Author Response · Authors · 2022-08-02
> **Responses to Reviewer 4H5f**
>
> - **About the sampling algorithm.**
>
> We add an algorithm section in Appendix A.3 to explain how the sampling step works. Please see our revised appendix file in **Supplementary Material**. The additional algorithm section is on page 2~3.
>
>
>
> - **About the figure layout and overall proofreading.**
>
> We will optimize the layout and proofread the paper in following revision to make our paper readable.
>
>
>
> - **About L116.**
>
> Sorry for the inaccurate statements that confused you. Your interpretation is correct.
>
>
>
> - **About the experiments in section 3.4.**
>
> As you suggested, we discard our encoder and directly incorporate the class label of MNIST sample as one-hot vector into $G_{\psi}(x_{t}, y, t)$ to learn a mean shift by filling the posterior mean gap, i.e., optimize $\psi$ by minimizing $E_{t,(x_{0},y),\epsilon}\bigg[ \lambda_{t} \big\| \epsilon - \epsilon_{\theta}(x_{t}, t) + \frac{\sqrt{\alpha_{t}} \sqrt{1-\bar\alpha_{t}}}{\beta_{t}} \cdot \Sigma_{\theta}(x_{t}, t) \cdot G_{\psi}(x_{t}, y, t) \big\|^{2} \bigg]$.
>
> Then we carry out the mixed sampling procedure described in section 3.4 and get the same results. Please see our revised appendix file in **Supplementary Material**. The supplementary experiment is in Appendix D.1, on page 4. It's not surprising we see some effect. The mean shift during critical-stage actually contains more crucial information to reconstruct the class information in samples than the other two stages.
>
>
>
> - **About the meaning of "one-step".**
>
> "the prediction of $x_{0}$ is done from $x_{t}$ in only one step" means that:
>
> Given a sample $x_{0}$ and a timestep $t$, we first calculate $x_{t}=\sqrt{\bar\alpha_{t}}x_{0}+\sqrt{1-\bar\alpha_{t}}\epsilon$.
>
> Then for pre-trained DPMs, we directly predict the noise in $x_{t}$ as $\hat{\epsilon} = \epsilon_{\theta}(x_{t}, t)$ and output $\hat x_0 = \frac{x_{t} - \sqrt{1 - \bar\alpha_{t}}\hat\epsilon}{\sqrt{\bar\alpha_{t}}}$.
>
> For our method, we directly predict the noise in $x_{t}$ as $\hat\epsilon = \epsilon_{\theta}(x_{t}, t) - \frac{\sqrt{\alpha_{t}} \sqrt{1-\bar\alpha_{t}}}{\beta_{t}} \cdot \Sigma_{\theta}(x_{t}, t) \cdot G_{\psi}(x_{t}, E_{\varphi}(x_{0}),t)$ and output $\hat x_0 = \frac{x_{t} - \sqrt{1 - \bar\alpha_{t}}\hat\epsilon}{\sqrt{\bar\alpha_{t}}}$.
>
> Through the comparison of $\hat x_0$ between these two methods, we want to show that our method:
>
> 1. actually fills the posterior mean gap so that it can output $\hat x_0$ significantly similar to $x_{0}$ for only one-step denoising, even for large $t$.
> 2. can generate high-quality samples with fewer sampling steps than pre-trained DPMs. After all, it can generate a plausible sample in only one step. That's why our method can be applied as an auxiliary booster of pre-trained DPMs to significantly improve the sample quality in few sampling steps, as shown in our unconditional sampling experiments (section 4.6).
>
>
>
> - **About the training time.**
>
> A fair and intuitive comparison of the training time between Diff-AE and our method is difficult. Unlike our method, the training of Diff-AE involves both diffusion denoising and representation learning, while ours resolves them separately. We only claim that our method needs less representation-relevant training times (i.e., the number of images used to train the representation learner, not time).
>
> Nevertheless, we still calculate the training time of different models on our devices. For saving time, we only re-train the pre-trained DPM and Diff-AE with 13M images and multiply their running time by 10.
>
> pre-trained DPM: **FFHQ128-130M**  248h
>
> our method based on above pre-trained DPM: **FFHQ128-130M-z512-22M**  36.4h
>
> Diff-AE: **FFHQ128-130M-z512**  307h
>
> We can find that 248+36.4<307, which means that our method actually takes less training time and performs better than Diff-AE. If someone has alreadly had a pre-trained DPM, our method must be the preferred option for representation learning, which will save much more time than retraining a DPM (36.4<307).

---

> > ### Comment · Reviewer_4H5f · 2022-08-09
> > **The authors have addressed my issues.**
> >
> > I vote for accepting this paper as I think that it proposes a great approach and presents an improvement in the performance and learning time.

---

### Official Review · Reviewer_jf2c · 2022-07-09

**Rating:** 5
**Confidence:** 3
**Soundness:** 2 fair
**Presentation:** 2 fair
**Contribution:** 2 fair

**Summary:**

The paper presents a new learning framework for diffusion probabilistic models. Unlike the most existing diffusion probabilistic models, which mainly embed the data into a predefined distribution through the Markov chain, the proposed framework aims to learn a robust representation feature through a pre-trained diffusion models. The authors also introduce a weighting scheme redesign function to further improve the representation learning in the diffusion models. Extensive experiments are conducted on multiple scenarios (e.g. FFHQ, CelebA, Bedroom).

**Questions:**

- What's the key difference between the proposed framework with the latest state-of-the-art diffusion autoencoders (Diff-AE) [21]. While the authors consider Diff-AE as a baseline, the two works seem to be similar. The authors should clearly explain what is the difference between them.
- Besides, The qualitative comparison to the Diff-AE baseline is missing, even after considering the results in the Appendix. What's more, the quantitative improvement in Tables 1, 2, and 3 is limited compared to the  Diff-AE baseline.
- It takes me a hard time buying the $x_0\to x_t$ in Figure 1. If I understand correctly, this should be a diffusion processing in general. Then, does this contain any pre-trained parameters?
- Could please the authors illuminate the difference between these results shown in Figure 5? While the authors claimed to "some stochastic variations", it seems to be very hard to capture such a conclusion.

**Limitations:**

Limitations are not discussed.

**Strengths And Weaknesses:**

### Strengths
- The task of representation learning via diffusion probabilistic models is novel and interesting.
- The overall idea and the approach towards addressing it seems reasonable.

### Weaknesses
- The paper is not well written and hard to read.
- The technical novelty is limited. Neither the designed network nor the theory are new.
- The qualitative results are poor, and the quantitative improvement is limited.

---

> ### Author Response · Authors · 2022-08-02
> **Responses to Reviewer jf2c**
>
> - **About the difference between our method and Diff-AE.**
>
> From the perspective of auto-encoder, both our method and Diff-AE employ an encoder to learn a representation $z$ from $x_0$. The key difference is on the decoder side. For Diff-AE, it trains a conditional DPM $\epsilon^c_\theta(x_t,t,z)$ from scratch to reconstruct $x_0$. For our method, we leverage a pre-trained unconditional DPM $\epsilon^u_\theta(x_t,t)$ and employ a gradient-estimator $G_\psi(x_t, z, t)$ to model the mean shift to fill the gap between the predicted posterior mean $\mu_\theta(x_t,t)$ (derived by $\epsilon^u_\theta(x_t, t)$) and the true posterior mean $\widetilde{\mu}_t(x_t,x_0)$. In this way, we can reconstruct $x_0$ using our gradient-estimator $G_\psi(x_t, z, t)$ based on pre-trained $\epsilon^u_\theta(x_t, t)$.
>
> Compared with Diff-AE, our method doesn't need to re-train a DPM from scratch, which is very time-consuming. Furthermore, by redesigning the weighting scheme of training objective, our method can learn richer representations from images.
>
>
>
> - **About the qualitative comparison and the imrpovements.**
>
> In autoencoding reconstruction contexts, both Diff-AE and our method achieve the MSE of $1e^{-5}$ order and it's difficult to detect macroscopic differences among samples. In unconditional sampling contexts, it's also difficult to perceive the sample quality of the models with different FID scores. They all generate plausible samples. Therefore we only list the quantative comparison to the Diff-AE baseline.
>
> In quantitative results, we think that the improvement of more than 0.5 on FID score should not be underestimated, especially for big and high-resolution datasets. Otherwise, our method only needs about $\frac{1}{5}$ and $\frac{1}{10}$ of the representation-relevant training times and time compared with Diff-AE. We think the improvements are not limited.
>
>
>
> - **About the confusion in Figure 1.**
>
> Sorry for the vague description in Figure-1 that confused you. Your interpretation is correct. $x_0 \rightarrow x_t$ means the forward process of DPM, i.e., $x_{t}=\sqrt{\bar\alpha_{t}}x_{0}+\sqrt{1-\bar\alpha_{t}}\epsilon$, which does not involve any pre-trained parameters.
>
>
>
> - **About the stochastic variations in samples.**
>
> We add an example in Appendix D.3 to illustrate the stochastic variation in our method. Please see our revised appendix file in **Supplementary Material**. It's on page 4~5.
>
> DDPM sampling method and DDIM sampling method (random $x_T$) always involve the stochastic variations, expressed in hair, skin and so on. With $x_T$ inferred from DDIM forward ODE, DDIM sampling method can almostly reconstruct the input image, without macroscopic difference. This also can be proved by the MSE metric in our **Autoencoding Reconstruction** experiments (section 4.2). DDIM sampling method with inferred $x_T$ achieves a much lower MSE than that  with random $x_T$, for both Diff-AE and our method.
>
>
>
>
> We sincerely hope you can reconsider the novelty and contributions of our method.

---

### Official Review · Reviewer_eiJZ · 2022-07-11

**Rating:** 5
**Confidence:** 5
**Soundness:** 4 excellent
**Presentation:** 4 excellent
**Contribution:** 2 fair

**Summary:**

This paper presents a method of learning representations out-of pretrained unconditional diffusion models. Different from DiffusionAE which learns the encoder and condition DPM together, this paper proposes to keep a pretrained unconditional diffusion models unchanged, and learn the gradient of classifier guidance which takes a latent variable as the additional input, and the latent variable in encoded by a learnable encoder. Empirical results show that this method leads to better reconstruction and few-shot conditional generation results compared to DiffusionAE with less training expense. Qualitatively the method learns meaningful representations, and it can also improve the sample quality of unconditional diffusion models with a learned latent DPM on the latent variables.

**Questions:**

- One baseline worthwhile to compare is DiffusionAE + classifier-free guidance. If the proposed method can beat this baseline, it will be more convincing that the improvement is given by the novel formulation $G_\psi$, instead of the guidance formulation which is not the contribution of this paper.

- For classifier(-free) guidance, it is known that results are better if the diffusion models are also conditional, i.e., the modified score is given by $\nabla_{x_t} [\log p(x_t|c) + \omega \log p(c|x_t)]$ instead of $\nabla_{x_t} [\log p(x_t) + \omega \log p(c|x_t)]$. I'm wondering if similar conclusion applies here. I.e., if you condition both the diffusion model and the classifier gradient $G_\psi$ with the latent variables $z$, whether you can get better performance than the current formulation.

**Limitations:**

No limitation and potential negative social impact are discussed .

**Strengths And Weaknesses:**

Strength:

- This paper is well written and easy to follow. It clearly states its connection with the previous close related work, DiffusionAE, and points out the advantage over the previous work.
- The proposed method is technically sound and naturally combines learning representation and classifier guidance.
- Various experiments are carried out and results are compared with baseline methods, making the method more convincing.

Weaknesses:
- IMO the novelty of this work is kind of limited. Compared to DiffusionAE, this work includes the classifier guidance and the latent codes are embedded into the learnable guidance term instead of the diffusion model, expect for which all other components remains the same as DiffusionAE, such as reversing DDIM sampler for latent code and learning latent DPM for z. Technically speaking I don't think there's a significant novel method proposed here.
- Compared to DiffusionAE, this method includes guidance term so it is not quite surprising to me that it can outperform DiffusionAE. It's not clear to me whether the improvement is due to the guidance mechanism or due to the novel formulation $G_\psi$.

---

> ### Author Response · Authors · 2022-08-02
> **Responses to Reviewer eiJZ**
>
> - **About Diff-AE with classifier-free guidance.**
>
> For a Diff-AE $\epsilon_\theta(x_t,t,z/\varnothing)$ trained by classifier-free method, i.e., randomly discard conditioning with some probability, we can use the pre-trained unconditional DPM $\epsilon^u_\theta(x_t,t)$ to simulate $\epsilon_\theta(x_t,t,\varnothing)$ and the trained Diff-AE $\epsilon^c_\theta(x_t,t,z)$ to simulate $\epsilon_\theta(x_t,t,z)$. In theory, this substitution at least performs not worse than that trained by classifier-free method. For the unconditional DPM with classifier guidance, we use modified diffusion score estimator $\epsilon^u_\theta(x_t,t)+(1+s)\cdot [\epsilon^c_\theta(x_t,t,z)-\epsilon^u_\theta(x_t,t)]$. Here we use $(1+s)$ because uncondtional DPM needs at least $1 \times$ classifier guidance to achieve conditional sampling. For the conditional DPM with classifier guidance, we use modified diffusion score estimator $\epsilon^c_\theta(x_t,t,z)+s\cdot[\epsilon^c_\theta(x_t,t,z)-\epsilon^u_\theta(x_t,t)]$, which is equivalent to above unconditional one.
>
> Moreover, we also adapt our method to classifier-guided form with the modified diffusion score estimator: $\epsilon_\theta(x_t,t)-(1+s)\cdot\frac{\sqrt{\alpha_t}\sqrt{1-\bar\alpha_t}}{\beta_t}\cdot\Sigma_\theta(x_t,t)\cdot G_\psi(x_t,E_\varphi(x_0),t)$. Then we perform the unconditional sampling on FFHQ-128 with DDIM (T=10) sampling method (same with section 4.6) for different strengths. The FID scores are shown in the following table:
>
> | Method  | s=0.0 | s=0.1 | s=0.2 | s=0.3 | s=0.4 | s=0.5 | s=1.0 | s=5.0 |
> | :-----: | ----- | ----- | ----- | ----- | ----- | ----- | ----- | ----- |
> | Diff-AE | 21.95 | 21.93 | 22.06 | 22.01 | 22.19 | 22.58 | 22.96 | 23.14 |
> |  ours   | 19.39 | 19.42 | 19.53 | 19.58 | 19.56 | 19.49 | 19.96 | 20.71 |
>
> As you can see, FID scores almost remain unchanged and large-strength classifier guidance will decrease FID scores.
>
> We think there are two reasons:
>
> 1. Classifier guidance are used to achieve the truncation-like effect, i.e., decrease the diversity of the samples while increasing the quality of each **individual** sample. The whole quality of samples, such as FID metric, will not necessarily improve, like the curve in Figure-4 of classifier-guided paper [1] and Figure-4 of classifier-free paper [2].
> 2. Our topic is totally different with those in classifier-guided [1] and classifier-free [2] paper. The classifiers they use are based on the class label of data, which is a kind of incomplete information of data. Therefore, reinforcing the strength of classifier guidance can compel samples to contain more information of class. But Diff-AE and our method use the classifiers that contains almost all information of data, i.e., $z$. Reinforcing the strength of classifier guidance is non-meaningful because they can have already reconstructed the data that $z$ corresponds to. We illustrate this by an experiment. Please see our revised appendix file in **Supplementary Material**. The supplementary experiment is in Appendix D.2, on page 4~5. Compared to the Figure-3 in classifier-guided [1] paper, where reinforcing the strength of classifier guidance extremely improve the sample quality, while in our topic, generated samples guided by different strengths are almost the same.
>
> So we think our improvements are given by the novel formulation $G_\psi$​ and redesigned weighting scheme, not the usage of classifier guidance. Also, classifier-guided and classifier-free methods introduce prior knowledge of data to fill the posterior mean gap (ends), while in our method, we learn the knowledge from data by filling the posterior mean gap (means). This is a novel contribution.
>
> - **About conditioning both the diffusion model and the gradient-estimator.**
>
> One of our main contributions is to use pre-trained DPMs for representation learning, so conditioning the DPMs with the representation $z$ means that we need to retrain both the DPM and gradient-estimator. In so doing, $G_\psi$ no longer models the gradient because its target is no longer the posterior mean gap. Actually there is no longer posterior mean gap between $\mu_t(x_t,x_0)$ and $\mu_\theta(x_t,t,z)$. The method transforms to Diff-AE.
>
> In practice, we train a model by optimizing  $L(\theta,\psi,\varphi)=E_{t,x_0,\epsilon}\bigg[\lambda_t\big\|\epsilon-\epsilon_\theta(x_t,t,E_\varphi(x_0))+\frac{\sqrt{\alpha_t}\sqrt{1-\bar\alpha_t}}{\beta_t}\cdot\Sigma_\theta(x_t,t)\cdot G_\psi(x_t,E_\varphi(x_0),t)\big\|^{2}\bigg]$.  Although we can perform DDPM sampling with $\epsilon_\theta(x_t,t,E_\varphi(x_0))-\frac{\sqrt{\alpha_t}\sqrt{1-\bar\alpha_t}}{\beta_t}\cdot\Sigma_\theta(x_t,t)\cdot G_\psi(x_t,E_\varphi(x_0),t)$, we cannot use $G_\psi$ as gradient for DDIM sampling, which makes the generated samples totally messy. Even it works, as we illustrate and explain above, the sample quality will be similar to our method.
>
> [1]: Diffusion Models Beat GANs on Image Synthesis
>
> [2]: Classifier-Free Diffusion Guidance

---

### Official Review · Reviewer_XByM · 2022-07-12

**Rating:** 7
**Confidence:** 3
**Soundness:** 4 excellent
**Presentation:** 4 excellent
**Contribution:** 4 excellent

**Summary:**

Whereas prior work (DiffAE) uses a conditional DPM as the decoder in an auto-encoder setup, this work attempts to leverage pretrained unconditional DPMs instead. The proposed formulation extends the classifier-guidance solution for latent autoencoder semantic codes z. However, instead of explicitly training classifiers p(z | x_t), taking its corresponding score, they propose to directly fit to the posterior mean gap of the pretrained, unsupervised, and frozen DPM. Results demonstrate higher quality reconstruction than DiffAE, with an overall simpler training setup.

**Questions:**

- See weaknesses
- How does the closeness between the pretrained DPM's domain and the target domain affect the efficacy of this method? If the pretrained DPM were from a completely unrelated domain, would this just mean that the posterior gap is larger and that the training time of this method would take much longer?

**Limitations:**

Limitations were not discussed. Example topics would include deep fakes etc.

**Strengths And Weaknesses:**

strengths
- paper was clearly written. Formulation was easy to follow
- prior works using classifier guidance had some issues regarding the scale of the guidance vector being used, but I believe the proposed formulation gets around that issue by fitting to the posterior mean gap instead.

weaknesses
- In section 3.4, it's not clear how the start and end of the critical stage are determined.
- L124 Except the latent code z (awkward phrasing)
- One thing that would've been nice to see is whether it's truly necessary to use another DPM for modeling the generative distribution $z$. DiffAE justifies their choice of a DPM by stating that a VAE's objective would have been difficult to tune, but perhaps a simple auto-encoder with a post-fit GMM over the latent vectors might suffice?

---

> ### Author Response · Authors · 2022-08-02
> **Responses to Reviewer XByM**
>
> - **About the start and end of the critical-stage.**
>
> We grid $1000$ timesteps with a step size of $50$ and perform grid-search for $(t_{1}, t_{2})$ paris to find the shortest critical-stage that can ensure high accuracy of conditional generation. For the DPM pre-trained on MNIST, it is $(400, 600)$. A shorter one will make the conditional guidance be eliminated by the stochasticity of reverse process. Different DPMs pre-trained on different datasets/domains may have different critical-stage, such as the critical-stage of $(350,700)$ for CIFAR-10 in our investigations.
>
> We originally worked without redesigning the weighting scheme, but found the training to be extremely unstable, resulting in slow/non convergence and poor performance. We supposed that the posterior mean gap in different stages may contain different levels/degrees of information and substantiated it through the experiments noted above. We then tried to redesign our weighting scheme to adjust the relative loss weighting of different timesteps, which is surprisingly well performed.
>
> Due to different critical-stage that different DPMs have, perhaps our method can be further improved by carefully redesigning the dataset/domain-dependent weighting scheme. We leave the investigations of this aspect as future work. Now in this project, our proposed weighting scheme applies and performs well for all pre-trained DPMs we use.
>
>
>
> - **About the writings.**
>
> Thanks for your suggestions and we will correct such awkward phrasing in following revision to make our paper readable.
>
>
>
> - **About modeling  the generative distribution $z$.**
>
> Actually we tried to model the generative distribution $z$ with a normalizing flow model, for which we just use a stack of affine coupling layers [1]. It can achieve similar performance with DPM, but its network is more complicated and training is more time-consuming than DPM. We also tried VAE to model the generative distribution $z$. However the training is unstable and the model cannot reach convergence. The sample quality is also under-performance. This may be due to its limited capacity or prior-hole problem. Perhaps deep hierarchical vaes, such as NVAE [2], can work.
>
> We try your idea that employs a simple auto-encoder with a post-fit GMM over the latent vectors. However we find that the parameter estimates of GMM is difficult. Many generated samples (GMM $\rightarrow$ $z$ $\rightarrow$ $G_\psi$ $\rightarrow$ $x_0$) are blurry or corrupted. We think this is because the generative distribution is complex and the gradient-estimator $G_\psi$ is sensitive to $z$, so that the auto-encoder+GMM cannot model it accurately and the variations of the latents of auto-encoder greatly affect the generated samples.
>
> Moreover, the DPMs we use to model the generative distribution are lightweight (stack of MLP layers) and easy to train. DPMs have advantages of stable training and excellent performance compared to other counterparts. We think that it's a good choice.
>
>
>
> - **About the domain problem.**
>
> We try your idea on MNIST. Specifically, we train a DPM on the images of 5 digital classes and use it to learn representations on the images of the other 5. It works well.
>
> However, when we use the DPM pre-trained on FFHQ to learn representations on LSUNBedroom, the training of our method is very unstable and the model cannot converge. The samples are totally messy.
>
> Despite the success on MNIST, we think that is because the simplicity of MNIST. Our method cannot handle the domain shift problem.
>
> In theory, the pre-trained DPM $\epsilon_{\theta}$ only recognize the noisy samples belonging to the distribution it is trained on. With some out-of-distribution $x_t$, the gap between $\epsilon$ and $\epsilon_{\theta}(x_{t}, t)$ will become non-meaningful (because $\epsilon_{\theta}(x_{t}, t)$ is non-meaningful) and our method will fail to learn useful representations from it.
>
> DiffusionCLIP [3] propose to fine-tune the pre-trained DPMs to adapt it to other domains, which means that we actually need a new $\epsilon_\theta$ for a new domain.
>
>
>
> [1]: Density estimation using real nvp
>
> [2]: NVAE: A deep hierarchical variational autoencoder
>
> [3]: DiffusionCLIP: Text-Guided Diffusion Models for Robust Image Manipulation

---

### Author Response · Authors · 2022-08-08
**We are sincerely looking forward to your reply.**

Dear reviewers,

we first thank you again for your valuable comments and suggestions. In the previous replies, we have tried our best to address your questions point by point.

We are sincerely looking forward to your reply to our responses and we are open to any discussions to improve our paper.

Best wishes!

The authors.

---

### Meta-Review · Area_Chair_BWPe · 2022-08-21

**Recommendation:** Accept
**Confidence:** Certain

**Metareview:**

This paper presents a new unsupervised learning method by making full use of pre-trained diffusion probabilistic models. Extensive experiments show that the proposed method can obtain an improvement in performance and learning time. Four reviewers voted for accepting the paper after the rebuttal and the discussion. All concerns raised by the reviewers have been well addressed by the authors. The AC agrees with the reviewers and recommends accepting the paper. Also, AC urges the authors to improve their paper by taking into account all the suggestions from reviewers.

**Award:**

No

---

### Decision · Program_Chairs · 2022-09-14

Accept